# Solution-grown BiI/BiI₃ van der Waals heterostructures for sensitive X-ray detection

Renzhong Zhuang[1,2], Songhua Cai [ORCID][3], Zengxia Mei[1], Huili Liang[1], Ningjiu Zhao[1], Haoran Mu[1], Wenzhi Yu[1,4], Yan Jiang[1], Jian Yuan[1], Shuping Lau [ORCID][3], Shiming Deng[5], Mingyue Han[1], Peng Jin [ORCID][6], Cailin Wang [ORCID][1], Guangyu Zhang [ORCID][1,4] ✉ & Shenghuang Lin [ORCID][1] ✉

X-ray detectors must be operated at minimal doses to reduce radiation health risks during X-ray security examination or medical inspection, therefore requiring high sensitivity and low detection limits. Although organolead trihalide perovskites have rapidly emerged as promising candidates for X-ray detection due to their low cost and remarkable performance, these materials threaten the safety of the human body and environment due to the presence of lead. Here we present the realization of highly sensitive X-ray detectors based on an environmentally friendly solution-grown thick BiI/BiI₃/BiI ($Bi_xI_y$) van der Waals heterostructure. The devices exhibit anisotropic X-ray detection response with a sensitivity up to $4.3 \times 10^4$ µC Gy⁻¹ cm⁻² and a detection limit as low as 34 nGy s⁻¹. At the same time, our $Bi_xI_y$ detectors demonstrate high environmental and hard radiation stabilities. Our work motivates the search for new van der Waals heterostructure classes to realize high-performance X-ray detectors and other optoelectronic devices without employing toxic elements.

High-sensitive X-ray detection requiring a low-dose rate is of particular importance to reduce the risks of cancer caused by repeated exposure to ionizing radiation in the fields of physical examination such as medical diagnosis and security inspection[1,2]. Therefore, it promotes the exploration of X-ray detectors to improve the sensitivity and reduce the detection limit. High-sensitivity and low detection limit require the X-ray detectors to possess high resistivity, high attenuation coefficient, low electron–hole formation energy ($\varepsilon_{pair}$), and excellent charge collection ability. Here, "high resistivity" results in the selection of materials with a large bandgap to reduce the temperature-induced carrier excitation. Whereas "low $\varepsilon_{pair}$" needs the target materials with a small bandgap to

generate more electron–hole pairs by a single X-ray photon. Therefore, a medium bandgap between 1.5 and 3.0 eV is considered appropriate to balance the $\varepsilon_{pair}$ and resistivity[3]. Nowadays, excellent semiconductors such as metal halide perovskites and CZT in forms of single crystal, polycrystalline or thick film with medium bandgap have been developed for high-sensitive room temperature X-ray detection. However, they are still limited by toxicity, stability, or cost[4–9].

Apart from the mentioned semiconductors, $BiI_3$ is also an alternatively promising material with a medium bandgap. As reported, $BiI_3$ is a 2D-layered semiconductor that belongs to $R\bar{3}$ –148 space group with a strongly anisotropic crystal structure

[1]Songshan Lake Materials Laboratory, 523808 Dongguan, Guangdong, P. R. China. [2]Fujian Provincial Key Laboratory of Welding Quality Intelligent Evaluation, Longyan University, Longyan, Fujian, P. R. China. [3]Department of Applied Physics, The Hong Kong Polytechnic University, Hunghom, Kowloon, Hong Kong, P. R. China. [4]Institute of Physics, Chinese Academy of Science, 100190 Beijing, P. R. China. [5]HAMAMATSU Photonics (China) Co., LTD., 100020 Beijing, P. R. China. [6]State Key Laboratory of Modern Optical Instrumentation, College of Optical Science and Engineering, Zhejiang University, Hangzhou, Zhejiang, China. ✉e-mail: gyzhang@sslab.org.cn; linshenghuang@sslab.org.cn

consists of I–Bi–I tri-layers stacked by weak van der Waals interactions[10,11]. Owing to the appropriate bandgap (1.67 eV), high density (5.8 g cm⁻³), high atomic number ($Z_{Bi}$ = 83, $Z_I$ = 53), and high resistivity ($10^8$–$10^{13}$ Ω cm), $BiI_3$ is attractive for hard radiation detection and has achieved sensitive X-ray detection recent years[12–25].

This article reports a van der Waals heterostructure of $Bi_xI_y$ that served as a high-sensitive X-ray detector. By analyzing the samples' cross-sectional images by aberration-corrected scanning transmission electron microscopy (ac-STEM), we found the obtained $Bi_xI_y$ is composed of thick $BiI_3$ layers (main) alternately stacked with new thin Bi-rich layers (minor) with a chemical formula of BiI. Namely, $Bi_xI_y$ presents a heterostructure formed by the stackings of BiI/$BiI_3$/BiI, which leads to a dual bandgap. Benefited from the heterostructure, $Bi_xI_y$ exhibits a higher X-ray sensitivity than $BiI_3$ single crystal. Moreover, like many halide perovskite single crystals, macrosize $Bi_xI_y$ can be grown using a low-cost, handy low-temperature solution method. These advantages together with other merits such as good X-ray attenuation efficiency and charge collection ability, high resistivity, environmentally friendly, good environmental and hard radiation stability make $Bi_xI_y$ be attractive as a potential competitor for high-sensitive room temperature X-ray detection.

## Results

### $Bi_xI_y$ grown in solution

The $Bi_xI_y$ were grown by a low-temperature solution technique, as shown in Fig. 1a. First, $Bi_2O_3$, $I_2$, and Au were dissolved in a mixed hydroiodic acid and ethanol solution to form precursor solutions. Excessive iodine is used to dissolve gold in hydroiodic acid fully. Then the solution refinement technique was employed to grow high-quality crystals[26]. Namely, the precursor solutions were pretreated by solvothermal in 1,4-butyrolactone for three times and then refined by hydrothermal for 1-time. $Bi^{3+}$ was reduced by $I^-$ with Au acted as a catalyst during the solution pretreatment; the chemical reactions in the solution were:

$$4H^+ + 4I^- + O_2 \uparrow \xrightarrow{\triangle} 2I_2 \uparrow + 2H_2O \tag{1}$$

$$I_2 + I^- \underset{\triangle}{\overset{\triangle}{\rightleftharpoons}} I_3^- \tag{2}$$

$$Bi^{3+} + 2I^- \underset{}{\overset{\triangle/Au}{\rightleftharpoons}} Bi^+ + I_2 \uparrow \tag{3}$$

The iodine diffused into 1,4-butyrolactone and made it darken. $Bi_xI_y$ with regular hexagonal shape and size up to $6 \times 6 \times 1$ mm³ were obtained from refined solution after 14 days of water bath growth at room temperature without any disturbance, as shown in Fig. 1b. More times pretreatment in 1,4-butyrolactone would promote massive nucleation. Smooth surfaces and Large flexible flakes of the grown $Bi_xI_y$ can be obtained by mechanical exfoliation, as shown in Fig. 1b and Supplementary Fig. 1. The morphology image (Fig. 1d) collected by scanning electron microscopy (SEM) reveals that the studied $Bi_xI_y$ flake possesses a high-quality exfoliated surface with no bubbles, holes, and inclusions. The morphology profile (Supplementary Fig. 1) of $Bi_xI_y$ flakes observed by atomic force microscopy (AFM) shows a layered structure. A step height of 0.66 nm was obtained and assigned to I–Bi–Bi–I monolayer (0.65 nm obtained by ac-STEM, see below), as shown in Supplementary Fig. 1.

$Bi_xI_y$ has the same X-ray diffraction (XRD) pattern (Supplementary Fig. 2) as $BiI_3$, but exhibits broader peaks and preferred orientation along [001], indicates a softer nature of $Bi_xI_y$. The inductively coupled plasma atomic emission spectra (ICP-AES) result (35.2 wt.% Bi of as-grown $Bi_xI_y$) confirms the chemical composition is $BiI_3$ (35.4 wt.% Bi in calculation). However, thermogravimetric (TG) and differential scanning calorimetry (DSC) analyses (Supplementary Fig. 3) show a different thermal behavior between $Bi_xI_y$ and $BiI_3$. The melting point of

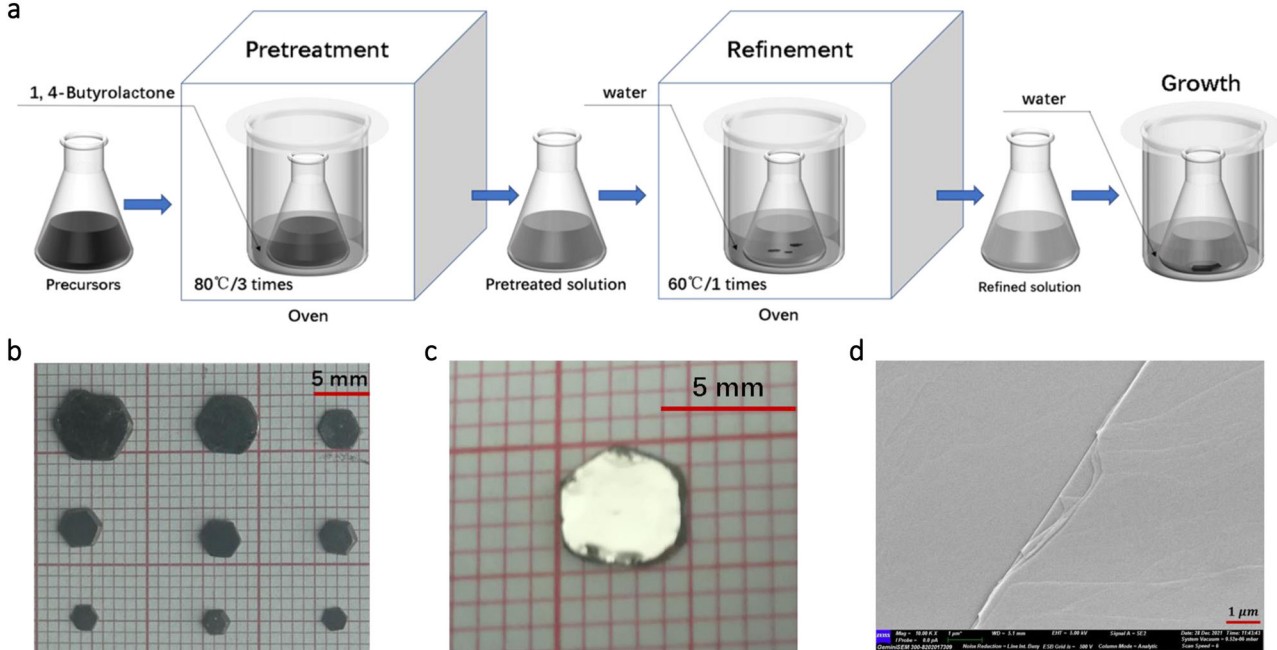

**Fig. 1 | Preparation and characterization of $Bi_xI_y$. a** A typical growth procedure for $Bi_xI_y$ includes solution pretreatment, solution refinement and crystal growth in room temperature water bath. **b** Optical image of as-grown crystals. **c** A typical as-grown crystal with the stripped surface. **d** SEM image of an exfoliated flake transferred onto a conductive tape.

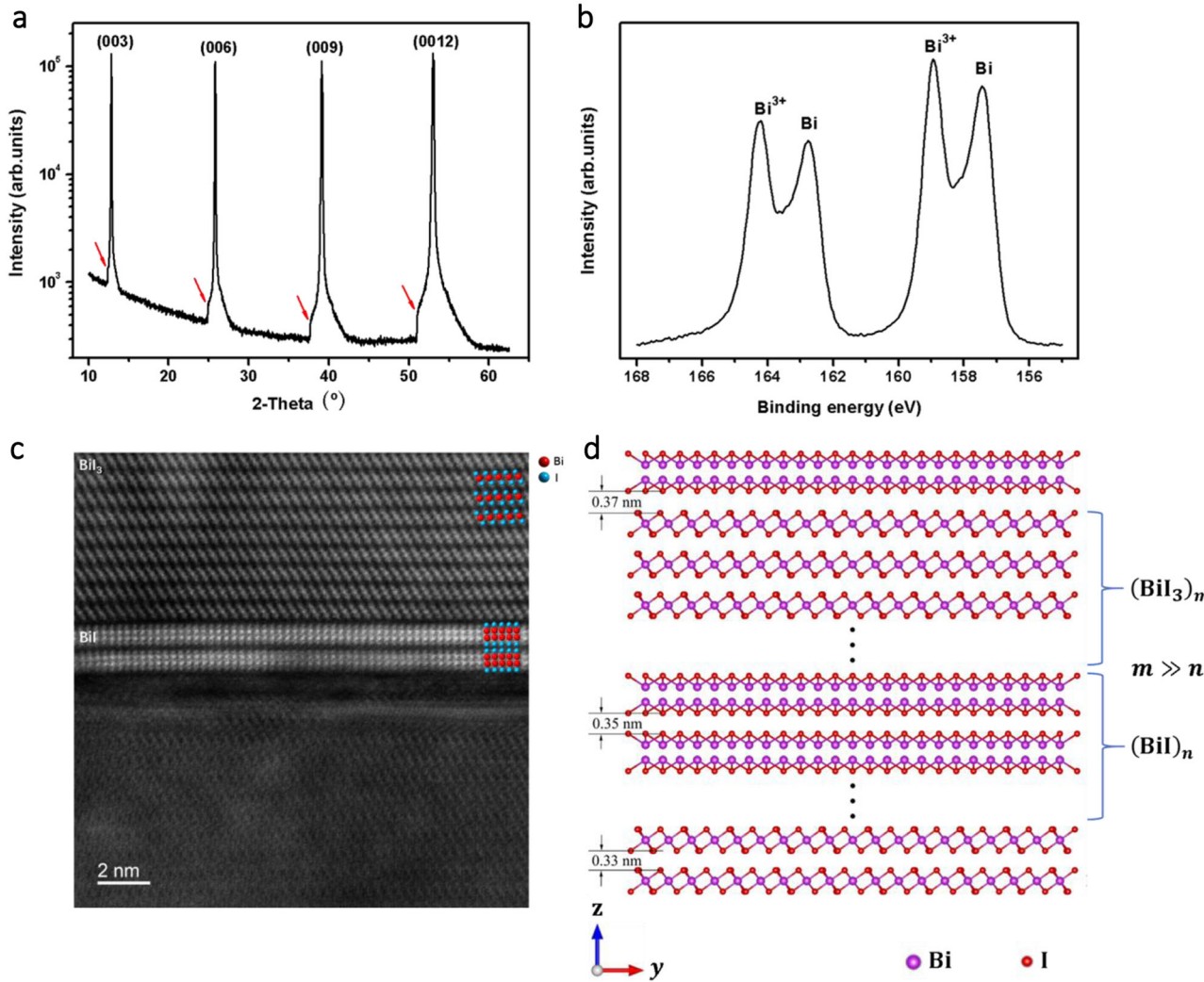

**Fig. 2 | Structure characterization of $Bi_xI_y$. a** (001) surface X-ray diffraction (XRD) pattern of $Bi_xI_y$, the red arrows point to a minor secondary diffraction of $Bi_xI_y$. **b** X-ray photoelectron spectroscopy (XPS) of Bi *4f* for $Bi_xI_y$. **c** High-angle annular dark-field scanning transmission electron microscope (HAADF-STEM) image of layered stacking of adjacent $BiI_3$ and BiI in $Bi_xI_y$. **d** Side view of $Bi_xI_y$ along [100] direction, where *m* and *n* are integer values. The thickness of BiI and $BiI_3$ layers is not controlled with atomic precision. The distance of $BiI_3$/BiI, BiI/BiI, and $BiI_3$/$BiI_3$ are 0.37 nm, 0.35 nm and 0.33 nm, respectively.

$Bi_xI_y$ (408 °C) is slightly different from $BiI_3$ (411 °C). $BiI_3$ exhibits two decomposition temperatures above melting point at 417 °C and 431 °C, respectively. However, which are not observed in $Bi_xI_y$. As the temperature exceeding 411 °C, $Bi_xI_y$ remains 46.2% of its original mass, on the other hand $BiI_3$ lost nearly all the mass. Meanwhile, no detectable mass loss is observed in the $Bi_xI_y$ even at a temperature of 300 °C, indicating its high thermal stability.

The stripped surface examined by XRD shows a heterostructure character of $Bi_xI_y$ with a major diffraction of $BiI_3$ (00k) and a regular minor secondary diffraction, as shown in Fig. 2a. The X-ray photoelectron spectroscopy (XPS) survey of Bi *4f* in the freshly stripped smooth surface of $Bi_xI_y$ shows the expected $Bi^{3+}$ peaks[27] at binding energies (BE) of 164.4 eV and 159.1 eV together with a distinct additional component shifted by 1.5 eV toward lower BE (Fig. 2b), assigned as $Bi^+$ (see below). The dramatic change of valance band BE from $BiI_3$ to $Bi_xI_y$ indicates a significant difference in the energy band between $BiI_3$ and the $Bi_xI_y$, as shown in Supplementary Fig. 4.

Then we used aberration-corrected STEM to observe the stacking sequence of $Bi_xI_y$ by a high-angle annular dark-field (HAADF) imaging mode, which provides directly interpretable Z-contrast images at the atomic level[28,29]. Supplementary Fig. 5 shows several cross-sectional STEM images of flakes with different thicknesses exfoliated from a grown crystal. As seen in Supplementary Fig. 5, the flakes show alternate stacking of thin bright layers and thick dark layers. The bright layers are assigned to a Bi-rich phase due to the Z-contrast HAADF image[28,29]. It should be noted here that the thickness of the exfoliated $Bi_xI_y$ is mainly dependent on the middle part of $BiI_3$. The STEM-EDS mapping results in Supplementary Fig. 5 also confirm a higher concentration of Bi in the bright layers. Detailed Z-contrast images of Bi-rich phase and the dark layers are shown in Supplementary Fig. 6. Bi-rich phase exhibits a layered van der Waals structure built by the stacking of I–Bi–Bi–I four atomic layers with a chemical composition of BiI. On the other hand, the dark layers have a $BiI_3$ structure characterized by the staking of I–Bi–I three atomic layers[23], which is consistent with the $BiI_3$ atomic structure model. $BiI_3$ and BiI layers are also held together by weak van der Waals force in the $Bi_xI_y$ structure, as shown in Fig. 2c. The STEM images clearly confirm the van der Waals heterostructure of $Bi_xI_y$ constructed by stacking of thick $BiI_3$ and thin BiI layers, as shown in Fig. 2d.

The BiI layers could be separated from the $Bi_xI_y$ by mechanical exfoliation. As a result, we successfully obtained the thinnest BiI film composed of seven I–Bi–Bi–I layers, as shown in Supplementary Fig. 7. Unlike other Bi-rich bismuth iodides such as $Bi_4I_4$ with 1D structure[30], the BiI we obtained is a member of 2D family and could be used to

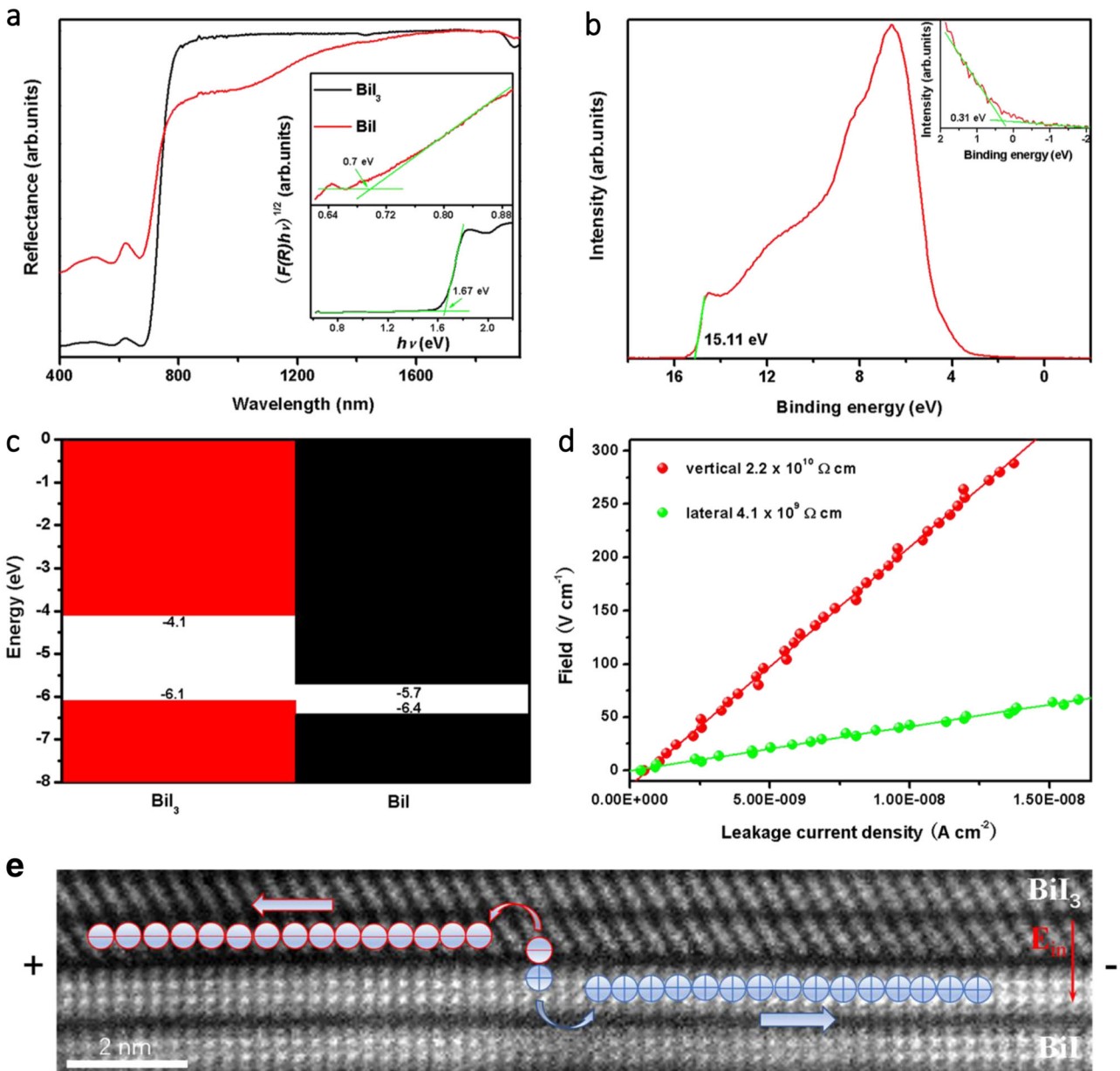

**Fig. 3 | Band structure and charge separation kinetics of $Bi_xI_y$. a** Diffuse reflectance spectra of commercial $BiI_3$ and $Bi_xI_y$. Inset: calculated optical bandgaps of $BiI_3$ layer and $BiI$ layer in the $Bi_xI_y$ using the Tauc method by assuming an indirect bandgap. As using the reflectance, the Tauc relation is: $(F(R)h\upsilon)^{1/2} = A(h\upsilon\text{-}E_g)$, where $A$ is a constant, $h$ the Planck constant, $\upsilon$ the photon frequency, $F(R) = (1-R)^2/2R$, $R$ the reflectance. The linear extrapolation (the green lines) in the absorption edge region of $(F(R)h\upsilon)^{1/2}$ versus $h\upsilon$ curve is used to determine the bandgap $E_g$. **b** UPS spectrum of $Bi_xI_y$, the linear extrapolation (the green line) is used to determine the cutoff energy (15.11 eV). Inset: linear extrapolation (the green lines) in the low-binding-energy region, the energy (0.31 eV) determined is used to calculate the valence band energy ($E_\upsilon$) as: $E_\upsilon = 21.22 - 15.11 + 0.31 = 6.42$ eV of BiI layer in the $Bi_xI_y$. **c** Band diagram of $Bi_xI_y$ exhibits a dual bandgap, the values of $E_c$ (conduction band energy) and $E_\upsilon$ (valence band energy) of $BiI_3$ are extracted from ref. [32]. **d** A typical leakage current-field relation of the Ag/$Bi_xI_y$/Ag devices tested by probe, Ag paste was used to fabricate the device, the red and green symbols are experimental data, the solid lines are linear fitting of the experimental data. **e** Excitation of electron–hole pairs separated at the $BiI_3$–BiI interface. The cross-sectional image is cut out from a STEM image and used as a schematic diagram. The red and blue circles represent electron and hole, respectively. The red and blue arrows illustrate their flow under bias.

build the blocks of van der Waals heterostructures with other 2D atomic crystals. Moreover, considering the versatility of 1D $Bi_4I_4$ in thermoelectric, topological insulator, and superconductivity[31], the 2D BiI may also exhibit similar properties.

## Dual bandgap of $Bi_xI_y$

The bandgap of the $Bi_xI_y$ was measured by UV–Vis–NIR diffuse reflectance spectroscopy (DRS) and shown in Fig. 3a. The reflectance of $Bi_xI_y$ exhibits an indirect band nature with a sharp increase at 670–810 nm, assigned to the thick $BiI_3$ layers, and a gentle rise after 810 nm,

assigned to BiI layers. Namely, $Bi_xI_y$ exhibits a dual bandgap. The dual bandgap of $Bi_xI_y$ is also confirmed by the absorption spectrum (Supplementary Fig. 8) of a typical grown $Bi_xI_y$. Based on the DRS result, the bandgap energy ($E_g$) of BiI layer was obtained by the Tauc method with 0.70 eV, dramatically lower than that of $BiI_3$ (1.67 eV) layer. The reduced bandgap of BiI promotes photo-responses up to 1800 nm in the $Bi_xI_y$, as shown in Supplementary Fig. 9.

Figure 3b shows ultraviolet photoelectron spectroscopy (UPS) of the $Bi_xI_y$, which exhibits different cutoff energy (15.11 eV) and the energy (0.31 eV) extracted from linear extrapolation in the

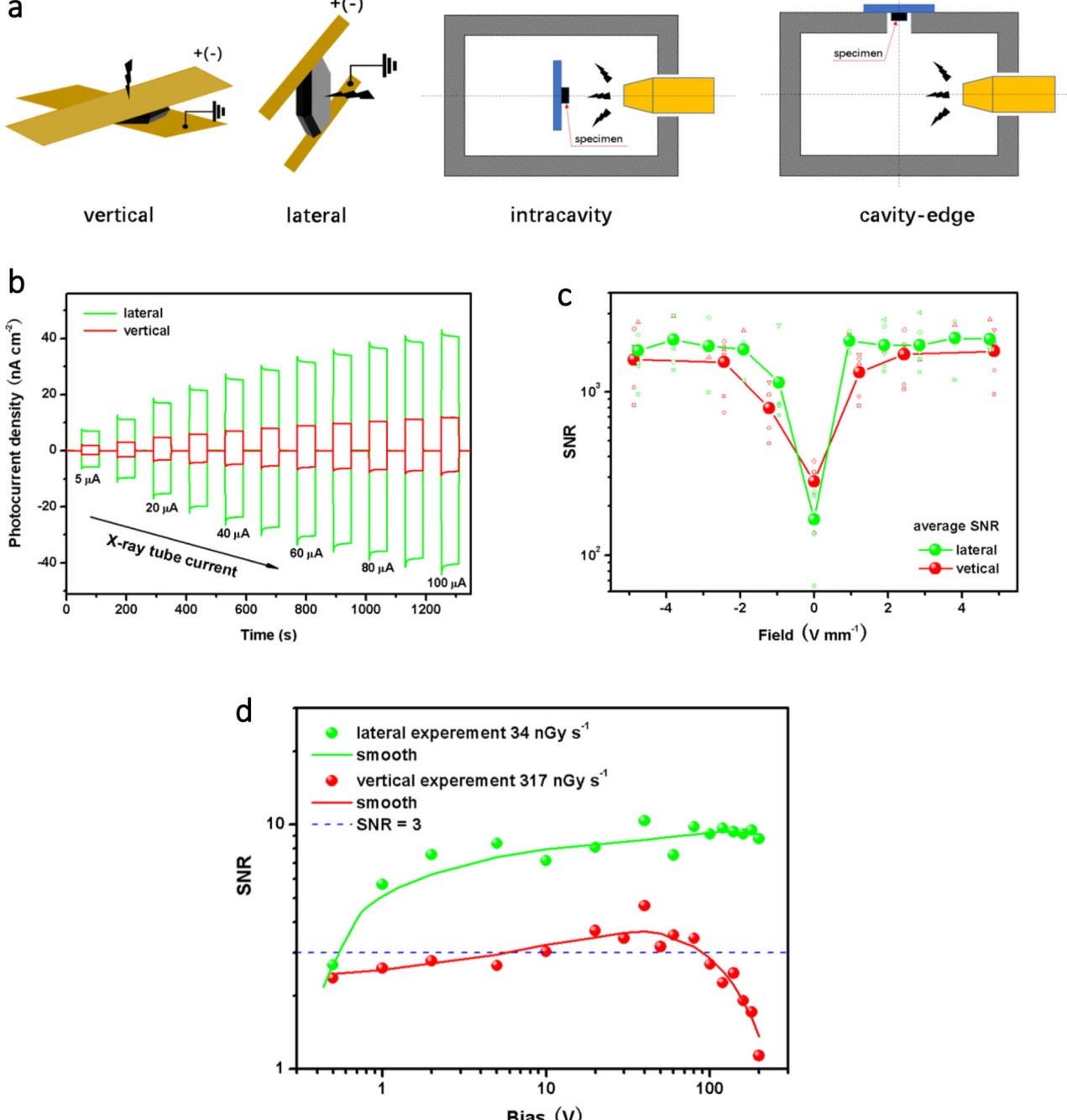

**Fig. 4 | Room temperature device performance. a** Illustration of X-ray detector (the Cu tapes cover the whole surface for full charge collection) and measurement configuration. **b** Anisotropic on/off X-ray responses of $Bi_xI_y$ at different dose rates measured by intracavity configuration under 1 V mm$^{-1}$ bias. **c** Anisotropic bias-dependent SNR (average from values under different dose rates with an X-ray tube current of 5–100 µA) measured by intracavity configuration. **d** Anisotropic bias-dependent SNR measured by cavity-edge configuration. The blue dotted line represents a SNR of 3, so the detection limits are 34 nGy s$^{-1}$ for lateral and 317 nGy s$^{-1}$ for vertical devices, respectively. The solid lines are smooth of the experimental data.

low-binding-energy region from which of $BiI_3$ (about 16.75 eV and 1.25 eV, respectively, see ref. [32] and Supplementary Fig. 4). As seen from Fig. 2d, the distance between $BiI_3$ and BiI layer (0.37 nm) is larger than which between $BiI_3$ layers (0.33 nm), resulting in rich of BiI in the stripped surface of $Bi_xI_y$. Therefore, the valence band energy ($E_v$) obtained by $E_v = 21.22 - 15.11 + 0.31 = 6.42$ eV is assigned to the BiI layer in the $Bi_xI_y$. The conduction band energy ($E_c$) of BiI layer is then calculated by $E_c = E_v + E_g = 5.72$ eV. The band diagram of $Bi_xI_y$ heterostructure with a dual-bandgap can be plotted using the above data, as shown in Fig. 3c. $Bi_xI_y$ exhibits large and anisotropy resistivities of

$4.1 \times 10^9$ Ω cm for lateral ($\mathbf{E} \perp c$) and $2.2 \times 10^{10}$ Ω cm for vertical direction ($\mathbf{E} \| c$), respectively, as shown in Fig. 3d. The resistivity of $Bi_xI_y$ is comparable to that of $BiI_3$ ($10^8$–$10^{13}$ Ω cm)[12–25], indicating a negligible effect of BiI layer with small bandgap (0.7 eV) to the resistivity of $Bi_xI_y$ due to its few numbers.

It can be deduced from the band diagram that electron injection from $BiI_3$ layer to BiI layer would happen at the $BiI_3$–BiI interface, which is confirmed by the transient absorption (TA) measurement (Supplementary Fig. 10). As seen from Supplementary Fig. 10, $BiI_3$ exhibits an absorption bleaching at 645–675 nm, but no bleaching was observed in

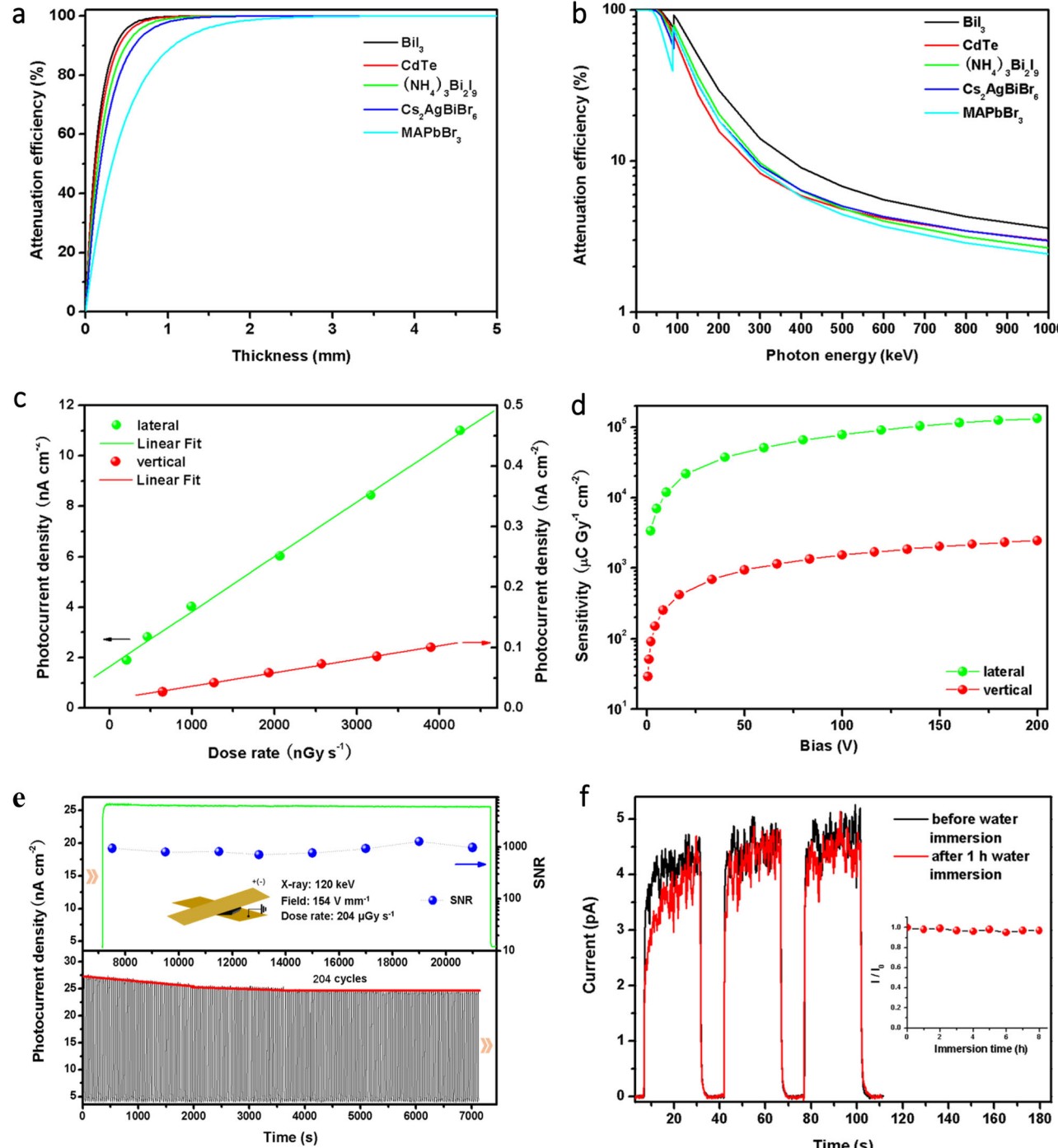

**Fig. 5 | X-ray attenuation, sensitivity, and stability of $Bi_xI_y$ detector at room temperature. a, b** Attenuation efficiencies of $BiI_3$, $(NH_4)_3Bi_2I_9$, $Cs_2AgBiBr_6$, $MAPbBr_3$ and CdTe semiconductors versus thickness to 50 keV X-ray photons (**a**) and photon energy (**b**). The curves were calculated by employing the NIST XCOM photon cross-section database[43]. **c** Anisotropic X-ray photocurrent densities at different dose rates measured by cavity-edge configuration under 1 V mm$^{-1}$ bias, the linear fittings (solid lines) are used to calculate the sensitivity. **d** Anisotropic X-ray sensitivities at different bias measured by cavity-edge configuration. **e** Device stability under repeated and continuous X-ray radiation. The black line is 204 repeated on/off response, followed by a response (green line) with a long continuous "on" time.

The red line illustrates a stable "on" current is achieved after a repeated on/off process. The blue symbols illustrate the fluctuation of device SNR during the long-time continuous X-ray radiation. Inset: a Cu/$Bi_xI_y$/Cu vertical device used for radiation stability test under 120 keV/204 uGy s$^{-1}$ X-ray radiation and 154 V mm$^{-1}$ field. **f** Device stability under humidity. The specimens, after water immersion, were dried by handkerchief tissues first and then repasted the Cu tapes for the following I−V tests. Inset: the X-ray response current changed with water immersion time up to 8 h, $I_0$ is the response current without water immersion. The absorption of the copper conductive tape was subtracted from all the data shown in (**a**−**f**).

the $Bi_xI_y$. Photocarriers generated by incident laser can form full-filled electrons in the conduction band of $BiI_3$, resulting in filled state bleaching. However, the bleaching disappeared in the $Bi_xI_y$ due to photoexcited electron transferring from $BiI_3$ to BiI layers at the $BiI_3$-BiI

interface instead of maintaining a filled state in the conduction band of $BiI_3$. The electron injection is also confirmed by a much weaker fluorescence around 700 nm of $Bi_xI_y$ than $BiI_3$, as shown in Supplementary Fig. 11. Due to the electron injection from $BiI_3$ layer to BiI layer, a built-

in field with a direction from $BiI_3$ to $BiI$ emerges at the $BiI_3$-$BiI$ interface, which would prompt the electron−hole separation under X-ray radiation, as shown in Fig. 3e. The separation of electrons and holes into different layers would weaken their recombination and promote charge collection.

## X-ray response with low detection limit

As mentioned above, the heterostructure of $Bi_xI_y$ has high resistivity and shows benefits for charge separation. Moreover, Bi bilayer is observed in the middle of I−Bi−Bi−I four atomic layers (Supplementary Fig. 6) and further confirmed by the Raman spectrum of bismuth (Supplementary Fig. 12), which is promising for ultrafast electron transport because of the ultrahigh electron mobility of bismuth[33]. Therefore, $Bi_xI_y$ is expected to have a good X-ray response with low noise.

Bulk ($2.1 \times 2.1 \times 0.4$ mm$^3$) X-ray detectors were fabricated with a device structure of Cu/ $Bi_xI_y$/Cu. The copper conductive tapes (0.06 mm thickness) were pasted on both sides of the surfaces parallel or perpendicular to (001) to fabricate the vertical or lateral devices, as shown in Fig. 4a. The devices were exposed to a source with X-ray photon energy up to 70 keV. Then X-ray induced photocurrents were measured by a normal intracavity direct radiation configuration and a cavity-edge leakage radiation configuration, as shown in Fig. 4a.

As seen from Fig. 4b, the photocurrents of lateral and vertical devices measured by intracavity configuration under various X-ray dose rates reveal clear on/off responses with anisotropy. A detailed on/off response of $Bi_xI_y$ (Supplementary Fig. 13) shows rise/fall times of 49/71 ms for lateral device and 74/98 ms for the vertical device under 1 V mm$^{-1}$ bias, which is comparable to the single crystal $BiI_3$ (110/120 ms) grown by physical vapor transport (PVT)[14] and the printable $MAPbI_3$ device (less than 50 ms) for imaging[34]. The signal-to-noise ratios (SNR) calculated from the on/off responses (method described in ref. [26].) show stable and large average values around 2000 of the lateral device and 1600 of the vertical device, under various X-ray dose rates and bias larger than 1 V mm$^{-1}$, as shown in Fig. 4c.

We also performed a leakage X-ray radiation measurement using a cavity-edge configuration (Fig. 4a) to examine the responses of the $Bi_xI_y$ detector in a simulated radiation leakage environment. As seen in Supplementary Fig. 14, the photocurrents induced by small leakage radiations still exhibit clear on/off responses in lateral and vertical devices. According to IUPAC standard, the dose rate with an SNR value of 3 is defined as the lowest detection limit at a given electric field. The lowest detection limit, representing the minimum X-ray dose rate used for inspection, is an important parameter relevant to health risk during X-ray security examinations or X-ray medical inspections[1,2]. As seen from Fig. 4d, the lowest detection limit of the lateral device achieved a very small value of 34 nGy s$^{-1}$, which is comparable to the excellent X-ray detectors with low detection limit[4–8]. The detection limit of the vertical device also achieves a small value of 317 nGy s$^{-1}$, much lower than that required for regular medical diagnostics (5.5 μGy s$^{-1}$)[35].

## X-ray sensitivity and stability

As mentioned above, thick $BiI_3$ layers constitute the main body of $Bi_xI_y$ with good X-ray radiation attenuation efficiency attributed to its high atomic number ($Z_{Bi} = 83$, $Z_I = 53$), and high density (5.8 g cm$^{-3}$). As seen from Fig. 5a, b, $BiI_3$ showed a much better X-ray attenuation efficiency than $MAPbBr_3$. For 50 keV hard X-ray, $BiI_3$ would attenuate 99.82% of the incident photons, while $MAPbBr_3$ 88.41% at 1 mm thickness. Therefore, high attenuation efficiency enables $Bi_xI_y$ to adequately absorb X-ray with reduced thickness, accelerating the charge collection. The charge collection ability of $Bi_xI_y$ characterized by a $\mu\tau$ product, where μ is the carrier mobility and τ the carrier lifetime, is derived by fitting the photoconductivity using Hecht equation[36], as shown in Supplementary Fig. 15. The $Bi_xI_y$ exhibits anisotropic $\mu\tau$ products of

$3.0 \times 10^{-3}$ cm$^2$ V$^{-1}$ (lateral) and $4.4 \times 10^{-5}$ cm$^2$ V$^{-1}$ (vertical) respectively. The corresponding lateral $\mu\tau$ products is comparable to that of perovskites with good X-ray detection properties[5,26,37]. The mobilities of $Bi_xI_y$ measured by the space charge-limited current (SCLC) method confirmed the strong anisotropic charge transport with values of 53 cm$^2$ V$^{-1}$ s$^{-1}$ (lateral) and 0.15 cm$^2$ V$^{-1}$ s$^{-1}$ (vertical), respectively, as shown in Supplementary Fig. 16. According to the electronic dimensionality theory[38], the electronic bands are more dispersive in the (001) plane of $Bi_xI_y$ owing to the strong in-plane chemical bonds interaction, while more localized perpendicular to the (001) plane induced by the weak out-of-plane van der Waals interaction. Therefore, $Bi_xI_y$ shows a better charge collection ability in the lateral direction.

Strong X-ray attenuation, high resistivity, good charge collection ability, and the resulting apparent responses at weak radiation indicate the $Bi_xI_y$ is highly sensitive to X-ray. The sensitivity of X-ray detectors is derived from the current-dose rate relations, as shown in Fig. 5c. The lateral device has much larger sensitivity than the vertical device, as shown in Fig. 5d. The obtained sensitivity of lateral device achieved a high value of $4.3 \times 10^4$ μC Gy$^{-1}$ cm$^{-2}$ at 24 V mm$^{-1}$ (50 V) bias, which is comparable to the newly reported high-sensitive detectors[37,39–42], shown in Supplementary Fig. 17. Moreover, the sensitivity of lateral device achieves nearly one order of magnitude higher than that of the lateral $BiI_3$ single crystal detector ($0.5 \times 10^4$ μC Gy$^{-1}$ cm$^{-2}$ at 20 V mm$^{-1}$)[19], confirmed the advantage of heterostructure of $Bi_xI_y$ for X-ray detection. However, the sensitivity of vertical device (253 μC Gy$^{-1}$ cm$^{-2}$ at 19.5 V mm$^{-1}$) is smaller than which of the vertical $BiI_3$ single crystal detector (660 μC Gy$^{-1}$ cm$^{-2}$ at 20 V mm$^{-1}$)[19]. A larger distance between $BiI_3$ and $BiI$ layers than which between $BiI_3$ layers leads to easier mechanical exfoliation of $Bi_xI_y$, however, harms the charge transport and then reduce the sensitivity of the vertical detector.

The $Bi_xI_y$ detector was exposed to repeated and continuous 120 keV X-ray (used for CT) with a dose rate of 204 μGy s$^{-1}$ to evaluate the anti-radiation stability. As seen from Fig. 5e, Stable X-ray photocurrent with a high SNR of around 1000 was observed after 204 circles repeated radiation and followed continuous radiation more than 4 h, confirms the highly stability of $Bi_xI_y$ detector under high energy X-ray radiation. Moreover, nearly unchanged photocurrent intensity (Fig. 5f) of the $Bi_xI_y$ detectors could be observed even after 8 h water (20 °C) immersion (Supplementary Fig. 18), confirms its highly environmental stability. High-sensitive and high-stable X-ray response of $Bi_xI_y$ detector offers its great prospects in real applications.

## Discussion

In summary, we developed a handy and scalable solution method to first grow the macrosize van der Waals heterostructure of $Bi_xI_y$ with regular shapes consisting of adjacent thick $BiI_3$ (main) and thin $BiI$ (minor) 2D layers. The $Bi_xI_y$ heterostructure X-ray detectors exhibit stable response and anisotropic properties at different crystal orientation. The lateral device realized a high sensitivity of $4.3 \times 10^4$ μC Gy$^{-1}$ cm$^{-2}$ with a very low detection limit of 34 nGy s$^{-1}$, meeting the demands of medical inspection to reduce the X-ray exposure to the human body. On the other hand, the $Bi_xI_y$ photodetectors are versatile and present a photo response ranging up to 1800 nm, revealing its potential for near-infrared detection. Generally speaking, our results inspire the exploration of van der Waals heterostructure materials for high-sensitive X-ray detection.

## Methods

### Precursor solution preparation

All the purchased chemical reagents except Au (99.999%) were of analytical reagent grade purity and used without further purification. Solution 1 was prepared by 5.5 g $Bi_2O_3$ it was dissolved in 20 ml 55% hydroiodic acid at room temperature. Solution 2 was prepared by 2 g

Au and 10 g $I_2$. They were dissolved in 5 ml 55% hydroiodic acid for 3 days at room temperature. Solution 3 was prepared by 10% hydroiodic acid mixed with ethanol in a volume ratio of 1:1. solution 1 and solution 2 were mixed and diluted by solution 3 to 40 ml. The prepared 40 ml diluted solution was used as a precursor solution.

## Solution pretreatment and refinement

The pretreatment and refinement procedures are schematically illustrated in Fig. 1a. Briefly, the obtained precursor solution was put into a 50 ml Φ20-mm conical flask. The flask was placed in a sealed beaker with 100 ml 1, 4-butyrolactone (GBL) for solvothermal treatment. Three times treatments are needed. The solvothermal treatment was performed in an 80 °C oven. After the first-time treatment (3–5 days), the solution was concentrated to 35 ml. The concentrated solution was diluted by solution 3 to 40 ml again for the second time solvothermal treatment. After the second time treatment, the solution was concentrated to 30 ml. The second time concentrated solution was diluted by ethanol to 35 ml for the third time solvothermal treatment. After the third time treatment, the solution was concentrated to 25 ml. The third time concentrated solution was then refined by hydrothermal treatment in a 60 °C oven for 3 days. Some small bulks with an irregular shape formed at the bottom of the conical flask after hydrothermal. The upper portion of the supernatant was then carefully transferred into another clear container to grow high-quality crystals.

## Crystal growth

The solution, after pretreatment and refinement, was then handled by water bath (Fig. 1a) at room temperature to grow crystals. More than 7 days of growth without disturbance is needed to obtain millimeter crystals. The obtained crystals were washed by ethanol one time and dichloromethane two times followed. Bulks after washing dried naturally in the air and used for the following material characterization, device preparation, and test.

## Characterization

Powder X-ray diffraction was performed on a D8-DISCOVER diffractometer with Cu Kα ($\lambda = 1.542$ Å) radiation. The X-ray Photoelectron Spectroscopy (XPS) and Ultraviolet Photoelectron Spectroscopy (UPS) were performed on a Thermo Fisher ESCALAB XI + photoelectron spectrometer. Freshly exfoliated surface after 30 s Ar ion sputtering was used for XPS and UPS measurements. Thermogravimetric analysis (TGA) was carried out under continuous nitrogen flow using a NETZSCH STA 449F3 thermal gravimetric analyzer. The sample was held on a platinum pan, and heated at a rate of 5 °C min$^{-1}$ up to 600 °C. AFM measurements were carried out in an Oxford Instruments Asylum Research Cypher S atomic force microscope with a contact mode. An IT500 scanning electron microscope (SEM) with a maximum 30 kV electron beam accelerating voltage was employed to observe the surface morphology of $Bi_xI_y$. STEM observations of the cross-section specimens were carried out in an aberration-corrected STEM microscope (Titan G2 60-300, Thermofisher equipped with a field emission gun) with 300 kV electron beam accelerating voltage. The probe convergence angle was 24.5 mrad, and the angular range of the HAADF detector was from 79.5 to 200 mrad. The cross-sectional TEM specimens were prepared by a dual-beam focused ion beam (FIB) nanofabrication platform (Helios 600i, Thermofisher). The UV–Vis–NIR diffuse reflectance spectroscopy (DRS) was measured by a HATACHI UH4150 spectrometer over the spectral range of 360–2000 nm. Room temperature photoluminescence and Raman spectra were collected by a Horiba LabRam HR Evolution microscopic confocal Raman spectrometer using a 6.8 mW, 532 nm CW Nd: YAG laser as an excitation source. The laser beam was focused to a spot size of about 0.7 μm in diameter. Transient absorption (TA) measurements were performed on a HARPIA-TA system (Light Conversion) at room temperature. A 1030 nm pulsed laser with 100 kHz repetition rate and 190 fs pulse duration was divided into two beams to generate pump laser and probe light, respectively. The pump laser of 480 nm was generated from an optical parametric amplifier system (OPA, Light Conversion) pumped by one beam of 1030 nm laser. The probe light was generated by exciting a sapphire plate by another beam of 1030 nm laser.

## Photo and X-ray response measurement

The device for photodetection was fabricated on the (001) surface of a $Bi_xI_y$ bulk, as shown in Supplementary Fig. 10. A pair of Ag electrodes with an interval of 0.5 mm was formed by painting Ag paste on a freshly exfoliated surface and then dried at 100 °C in the air. The area between Ag electrodes formed the light absorption area of the photodetector. The I–V characteristic under ambient light and infrared irradiation was measured by a KEITHLEY 2450 source meter. A YSL SC-PRO 7 supercontinuum source was used to generate CW infrared laser. Devices with Cu tape pasted on a pair of adjacent (100) or (001) surfaces formed the Cu/ $Bi_xI_y$/Cu structure (Fig. 3a), and were used for X-ray detection measurements. The X-ray detection performance was measured in a Pb cavity for intracavity mode and in a Φ 5 mm hole on the side of the cavity with a light-proof cover for cavity-edge mode, as shown in Supplementary Fig. 13. A commercially available MOXTEX MagPro Mini-X tube with a tungsten target and 12 W maximum power output was used as the X-ray source. The X-ray tube was operated with a constant 50 kV voltage. The total X-ray dose was modulated by changing the current of the X-ray tube. The radiation dose rate was calibrated using a Radical ion chamber dosimeter. The X-ray photocurrent was measured by a KEITHLEY 2636B source meter. For the anti-radiation test, a 150 kV HAMAMATSU/L12161-07 microfocus X-ray source with 75 W maximum power was used.

## Data availability

Relevant data supporting the key findings of this study are available within the article, the Supplementary Information file and the Source Data file. All raw data generated during the current study are available from the corresponding authors upon request. Source data are provided with this paper.

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

## Acknowledgements

S.H. Lin acknowledges the support from the National Key R&D Program of China (No. 2021YFA1202902), Guangdong Basic and Applied Basic Research Foundation (No. 2021B1515120034), and Songshan Lake Materials Laboratory (No. Y0D1051F211). S.H. Cai acknowledges support from the Hong Kong Polytechnic University (No. 1-BQ96), the General Research Fund (No. 15306021) from the Hong Kong Research Grant Council, the Research Grants Council of Hong Kong (No. C5029-18E), and the open subject of National Laboratory of Solid-State Microstructures, Nanjing University (M34001). R.Z. Zhuang acknowledges the prophase funding support from Longyan University. The authors thank Yu. Lin at the library of Longyan University for the literature retrieval service.

## Author contributions

R.Z. conceived the idea, and S.L. supervised the project. R.Z. and S.L. designed the experiments. R.Z. carried out material preparation and characterizations, device fabrication, and X-ray performance characterizations. S.C. and S.P.L. performed the STEM measurements. Z.M. and H.L. set up the measurement facilities for the X-ray detector. R.Z., S.D., Y.J., C.W., and M.H. performed the anti-radiation and humidity stability test. R.Z. and P.J. calibrated the radiation dose rate. N.Z. performed Transient absorption measurements. H.M., W.Y., and J.Y. set up the measurement facilities and performed ambient light and infrared response spectrum measurements. R.Z., S.L., and G.Z. wrote the manuscript with input from all authors.

## Competing interests

The authors declare no competing interests.
