## [Peer Review File · Nature Communications]

Solution-grown Bi/Bi₃ van der Waals heterostructures for sensitive X-ray detectionREVIEWER COMMENTS

Reviewer #1 (Remarks to the Author):

This paper introduces a new heterojunction detector, but there are many problems in the description and characterization of the materials.

1. It is confusing that the authors used Au in the process of growing single crystals. In particular, there are some problems with the chemical equations. In general, Bi^{3+} is not easily oxidized by Au. And these two equations look more like Bi^{3+} and I^- reactions catalyzed by Au. It is recommended to re-write these reaction equations more rigorously, taking into account the significant effects of the acid solution, oxygen, and other substances.

2. In Figure 1f, the description of the picture is missing.

3. I am very curious as to what is the driving force behind the formation of n layers of BiI and how the authors ensure that m and n are long-range ordered, i.e., BiI_3 and BiI will form ordered single crystals instead of m and n being chaotic values. The authors were unable to confirm it from XRD, and electron diffraction could only account for short-range alignments. It can be seen from the STEM in Figure S5 that the thickness of the BiI_3 layer is inconsistent, which is obviously not a single crystal of Bi_xI_y , but more like there are some plane defects in the single crystal of BiI_3 .

4. We can do peak separation and quantitative analysis through XPS data, Bi^+ has a similar number of atoms as Bi^{3+} , and the thermogravimetric data can also illustrate this point. This is inconsistent with the author's analysis that the number of Bi^{3+} atoms is far more than that of Bi^+ . And XRD does not show BiI-related peaks, only consistent with BiI_3 , which is both unreasonable and contradictory to the above conclusion. Furthermore, peak broadening cannot be seen in Fig. 1c, requiring an enlarged view. Faced with the phenomenon of peak broadening, in addition to the softer nature mentioned by the author, do we need to consider the Scherrer effect?

5. Is the bandgap measured in the article (0.70 eV, too small) suitable for X-ray detection? The author also mentioned that this value is lower than 1.5 eV, the default minimum value suitable for X-ray detectors. At such a small bandgap, why does it not exhibit a large leakage current? Especially from Figure S11, it can be seen that the photocurrent has prominent rising edges, falling edges, and the drift phenomenon, which proves that the sample is likely to have many defects.

6. The photoresponse in Figure 3c is very nonlinear, why? Will this affect the fitting process of the $\mu\tau$ value? Is the abscissa in Figure 3d wrong? It should not be the electric field strength.

Reviewer #2 (Remarks to the Author):

Zhaung et al reported Van der Waals heterostructure $\text{Bi}_x\text{I}_y/\text{BiI}_3$ device for low detection X-ray detector. The X-ray device has anisotropic transport in lateral and vertical direction which results in X-ray response. The X-ray device a low detection limit and high SNR with good stability. However, there still have many questions about this MS which require major revision as below.

In Figure 2, the authors use DRS to characterize the bandgap for heterojunction Bi_xI_y materials which will benefit the e-h pair collection. However, why the heterojunction material have huge bandgap decrease com 1.67 eV to 0.7 eV is not clear discuss and understand in the main text.

From Figure 2c, the heterojunction band alignment for BiI_3 and Bi_xI_y indicates the CB different is much smaller than VB between Bi_xI_y and BiI_3 materials. However, when the e-h pairs generated by X-ray, the e-h pair separation flow in fig 1e is not match with Fig 2c and is confusing. Based on Band alignment in Fig 2c, how the e-h pairs generated and transport in the band alignment?

In Fig 2d, the authors used calculated $\mu\tau$ products to understand the charge transport between lateral and vertical. However, it might be due to the applied bias is not high enough for thick crystal (or vertical direction). I would suggest the author use SCLC measure with higher applied bias to further confirm the results.

In Fig 3, the authors test the X-ray device performance at ambient condition. It's clear shows the difference of the X-ray response, SNR and detection limit for lateral and vertical devices. However, all of these device response time is in tens of seconds time regime. Why the device response time is so slow? The authors should have a detail discuss and comparison with reported other X-ray material candidates.

From fig 4f, the device response before and after water immersion is very encouraging, however, the crystal color changed after water immersion (or hydrolysis) and not quite the same in fig 1b. The authors should carefully characterized the material before and after water immersion in order to claim the material is not effected by water.

Also, from many device response, why some of the devices have baseline drifting issue and some not, why does these happen to the same material and devices?

Some minor correction:

(1) Some fig labeling and ordering in the main text is wrong, such as page 9 line 188. The fig 2e should be fig 2c. Fig2d is after fig 2e. Same for Fig 3. Please correct these issue through the whole MS.

(2) In Fig 2d, what does the parallel and vertical E means? It's confusing and not clear to reader. I would suggest the authors add a cartoon in the Fig 2d and clearly describe in the MS.

(3) Fig 4c and 4e are missing the right axis label

Reviewer #3 (Remarks to the Author):

As the first comment, I have to acknowledge the idea of BiI/BiI₃/BiI layered sample for use as an X-ray detector.

Besides this interesting idea, the characterization is lacking seriously.

Besides concerns on the electrical characterization, I encourage the authors to use neutral scientific language.

I.e champion material, ultra-high....are used everywhere. I know, these days, it is common to use superlatives in a scientific text, but this is a really bad practice. We have to show good scientific results - we are not selling on a marketplace.....

For the electrical characterization, I see several contradictions:

*) the authors report high resistivities for 0.7eV Bandgap material?! These high values are unphysical and are most likely caused by bad electrical contacts - not surprising for a Cu tape as a contact.

*) the authors report on extremely high sensitivities. What you see here is photoconductive gain. Very typical for a symmetric device. The IV of such a device has to be linear (not seen in the semi-log plot in Fig 2d). For such devices, you can get any sensitivity by applying a higher bias.

*) $\mu\tau$ product cannot be obtained from a symmetric device. I saw this approach in many papers but is an incorrect interpretation. The $\mu\tau E$ product defines the Schubweg of the charge carriers. At a certain bias, the Schubweg equals the sample thickness. For such an experiment you need ASYMMETRIC contacts (or at least one blocking). For examples. Take as reference a commercial CdTe photodiode. There you'll see sensitivity in the range of 3000-400 $\mu\text{C}/\text{Gscm}^2$ clearly saturating and exactly fitting the theoretical limit given by the Klein rule! Every value higher than this points to a leaky diode or photoconductive gain. Sorry to say, your treatment is not correct. I'm

picky here, as during my time in the industry I had to do such tests regularly.

***) Detection limit. Has to be measured at the same bias as the photocurrent. You cannot compare a 0V with the PC under bias. The correct treatment is to use a spectrum analyzer and determine the spectral noise density and from that, you can calculate the SNR.**

In summary. You are showing a highly interesting material system, but the characterization as an X-ray detector is lacking seriously.

Reviewer #4 (Remarks to the Author):

In the current work, Zhuang R. et. al demonstrate the synthesis of novel heterostructure BiI₃, which consists of sequential layers of BiI₃ and BiI, and based on that heterostructure the X-ray detector prototype. The authors did impressive work on the synthesis and the structural characterization of the BiI₃ material, clearly demonstrating its heterostructure form and well describing the novel solution-grown process. Optical and X-ray detection characterization was done partially correctly, proving high sensitivity to X-ray, low lowest detection limit, and reasonable stability. Overall the proposed BiI₃ heterostructure seems to be a promising material for X-ray detection. However, there are several major concerns, if unresolved, questioning the value of this work. The first concern is whether BiI₃ heterostructure is much better in terms of X-ray detection properties than a simple BiI₃ single crystal, which is pretty known in the literature as a good gamma and an X-ray detector, as the authors cite in ref 22-36. The second problem is the correctness of the BiI₃ heterostructure bandgap determination. Thus, I would recommend this manuscript for publication in Nature Communication only when all major and minor concerns are addressed, which are listed in detail below.

Major remarks

1. For a clear demonstration of this work advance, absolutely necessary is the direct comparison of X-ray detection performance between the fabricated BiI₃ heterostructure and similar size BiI₃ single crystal. While the authors provide in Supplementary Information a comparison of X-ray sensitivities (Fig. 15S), noting the sensitivity of BiI₃, it is hard to compare this value with the current work, since for BiI₃ reference isn't even stated. For a fair comparison, I suggest to the authors synthesize their own BiI₃ SC or utilize commercial ones, which the authors already used in work, fabricate the detector devices, and perform similar X-ray detection characterization. It is important to compare resistivity, mobility-lifetime, X-ray sensitivity, and lowest detection limit, to prove (or disprove) the advances achieved by the BiI₃ heterostructure. In the current state of the manuscript, only resistivity is compared properly (it is similar to the fabricated BiI₃ heterostructure and previously reported BiI₃), which definitely isn't a comprehensive comparison.

2. In the manuscript the only major physical properties, which is shown to be sufficiently different from BiI₃, is the bandgap. While the authors stated correctly, that a lower bandgap is beneficial for X-ray detection in terms of lower e-h creation energy, the bandgap values seem to be only estimated for BiI part, but not the whole heterostructure. For correct bandgap estimation of mixture compound, please look (DOI: 10.1021/acs.jpcclett.8b02892). It looks like there are separate bandgap values for different parts of the heterostructure for BiI and BiI₃ parts, and the last isn't changed compared to commercial BiI₃. As the authors stated, BiI contribution to chemical composition is tiny compared to BiI₃. Thus, X-ray are mostly attenuation in BiI₃ layers, and the effect of the low BiI bandgap on the e-h pairs creation energy is doubtful. This statement should be either clarified or removed from the manuscript. Despite that, I still think that heterostructure could have beneficial for charge transport properties

compared to BiI3, which is still to be proved, as it is stated in 1st comment.

Minor remarks

1. As being mentioned above, BiI3 material is well-known for radiation detection. While the authors mentioned a lot of perovskite references in the introduction, BiI3 research works are mentioned only in the context of resistivity. I suggest adding references on BiI3 detectors in the introduction, stating that this work is advanced compared to other BiI3 reports.

2. Please add the caption for Figure 1f.

3. I wouldn't use a double axis for Figure 4c, since it makes it difficult to compare two different directions. Instead, I would suggest making one axis with a log scale.

4. In the inset of Figure 2a, where the authors report the bandgap for BiIy (probably to BiI, as mentioned above), the authors may consider changing the Y-axis power from $\frac{1}{2}$ to 2, since the first is used for direct semiconductors, while the last is for indirect one. From reflectance dependence on main Figure 2a, it seems that BiI part has indirect semiconductor behavior.

Response to the reviewer's comments

Reviewer #1:

Reviewer #1 (Remarks to the Author):

This paper introduces a new heterojunction detector, but there are many problems in the description and characterization of the materials.

Response: Thanks very much for the reviewer's comments. We have revised the description and characterization of the materials according to the great comments of the reviewer in the **revised manuscript**.

1. It is confusing that the authors used Au in the process of growing single crystals. In particular, there are some problems with the chemical equations. In general, Bi³⁺ is not easily oxidized by Au. And these two equations look more like Bi³⁺ and I⁻ reactions catalyzed by Au. It is recommended to re-write these reaction equations more rigorously, taking into account the significant effects of the acid solution, oxygen, and other substances.

Response:

Thanks very much for the excellent suggestion. We found it is hardly to obtain large crystal in the solution without Au. The precipitants were fine powder BiI₃ without Bi⁺, as shown in the following figure. We know Au acted as catalyst during reaction and re-wrote the major reaction equations in the **revised manuscript** according to the reviewer's suggestion as follows:

XPS spectrum of Bi 4f for Bi_xI_y and BiI_3 .

2. In Figure 1f, the description of the picture is missing.

Response: Yes, it has been added in the **revised manuscript**.

3. I am very curious as to what is the driving force behind the formation of n layers of BiI and how the authors ensure that m and n are long-range ordered, i.e., BiI_3 and BiI will form ordered single crystals instead of m and n being chaotic values. The authors were unable to confirm it from XRD, and electron diffraction could only account for short-range alignments. It can be seen from the STEM in Figure S5 that the thickness of the BiI_3 layer is inconsistent, which is obviously not a single crystal of Bi_xI_y , but more like there are some plane defects in the single crystal of BiI_3 .

Response: Thanks for your good comments. We apologize for the unclear description of Bi_xI_y structure. Yes, m and n are not long-range ordered due to a larger entropy is preferred. Therefore, we failed to afford a crystal structure and used 'heterostructure' to describe the bulk Bi_xI_y instead of 'crystal' in the manuscript. However, we believe that it is not accurate to describe Bi_xI_y as a plane defect structure of BiI_3 single crystal because it cannot be derived from the disarrangements of the atomic planes of BiI_3 . Bi_xI_y is constructed by the van der Waals stacking of two different 2D component, which is clearly different from BiI_3 single crystal, therefore we describe it as 'van der Waals heterostructure'.

We added a description in Fig. 1f in the revised manuscript as "f, Side view of Bi_xI_y along [100] direction, m and n are chaotic values."

4. We can do peak separation and quantitative analysis through XPS data, Bi⁺ has a similar number of atoms as Bi³⁺, and the thermogravimetric data can also illustrate this point. This is inconsistent with the author's analysis that the number of Bi³⁺ atoms is far more than that of Bi⁺. And XRD does not show BiI-related peaks, only consistent with BiI₃, which is both unreasonable and contradictory to the above conclusion. Furthermore, peak broadening cannot be seen in Fig. 1c, requiring an enlarged view. Faced with the phenomenon of peak broadening, in addition to the softer nature mentioned by the author, do we need to consider the Scherrer effect?

Response: Great question, we know that Bi⁺ seems has a similar number of atoms as Bi³⁺ from the XPS data. However, the XPS signal gives atomic distribution of only several atomic layers of the surface. As seen from Fig. 1f, the distance between BiI₃ and BiI layers (0.37 nm) is larger than which between BiI₃ layers (0.33 nm). Therefore, the stripped surfaces trend to be rich of BiI. Since Bi_xI_y is a heterostructure rather than a simple mix of BiI and BiI₃, it cannot draw a conclusion from the TG analysis that the substances left after heating Bi_xI_y is BiI. We can also see from an added DSC analysis in Fig. S3 in the revised manuscript that Bi_xI_y exhibits a different thermal decomposition behavior from BiI₃. Bi_xI_y has only one melting point. Therefore, XPS and thermogravimetric data do not contradict to the conclusion derived from ICP and XRD. Due to the small number of BiI, we did not observe the BiI-related peaks from powder XRD. However, we can observe a minor secondary diffraction belongs to BiI in the (001) surface XRD pattern of Bi_xI_y using a log I - 2θ plot (Fig. 1c in the revised manuscript). We found Bi_xI_y is easy to cleave and exhibits some extent of malleable like metals during grinding, that is the softer nature and leading to preferred orientation along [001]. The peak broadening is more likely derived from the disorder stacking of Bi_xI_y rather than Scherrer effect.

The enlarged view which show the peak broadening is provided in Fig. S2 in the revised manuscript.

5. Is the bandgap measured in the article (0.70 eV, too small) suitable for X-ray detection? The author also mentioned that this value is lower than 1.5 eV, the default minimum value suitable for X-ray detectors. At such a small bandgap, why does it not exhibit a large leakage current? Especially from Figure S11, it can be seen that the photocurrent has prominent rising edges, falling edges, and the drift phenomenon, which proves that the sample is likely to have many defects.

Response: Good question, we know that single component semiconductors with small bandgap (< 1.5 eV) will exhibit large leakage current due to thermal excitation and are considered unsuitable for x-ray detection. However, Bi_xI_y is not a single component semiconductor due to its heterostructure. We made a mistake to take the bandgap of BiI (0.70 eV) as the bandgap of Bi_xI_y and corrected as "dual bandgap" in the revised manuscript. The main BiI₃ part of

Bi_xI_y has a medium bandgap of 1.67 eV, which leads to a low leakage current. Although BiI has a relatively high carrier concentration due to its small bandgap, it only contributes a small fraction of leakage current of Bi_xI_y as a minor part. Therefore, Bi_xI_y does not exhibit large leakage current. The visible and infrared photocurrent tests using the contact configuration shown in Fig. S9 in the revised manuscript are used to search the response limit of long wavelength. However, which are sensitive to many surface defects such as layer distortion, chipping and crack caused by stripping, surface oxidation, moisture and so on, which lead to the prolonged rising/falling edges and the drift phenomenon of photocurrent.

6. The photo response in Figure 3c is very nonlinear, why? Will this affect the fitting process of the $\mu\tau$ value? Is the abscissa in Figure 3d wrong? It should not be the electric field strength.

Response: Thanks for the question. The signal-to-noise ratios (SNR) in Fig. 3c is calculated from $I_{\text{signal}}/I_{\text{noise}}$, which increase as the bias increasing at first, indicating a nearly unchanged photo current noise under a low bias. As the bias continues to increase ($>1 \text{ V mm}^{-1}$), the SNR show nearly stable average values, indicating the photocurrent and its noise increase nearly in the same proportion. The noise will not affect the fitting process of the $\mu\tau$ value due to the SNR values are very large (around 2000 of the lateral device and 1600 of the vertical device respectively).

Yes, we used bias (V) in the new abscissa in the Fig. 3d in revised manuscript.

Reviewer #2:

Zhaung et al reported Van der Waals heterostructure $\text{Bi}_x\text{I}_y/\text{BiI}_3$ device for low detection X-ray detector. The X-ray device has anisotropic transport in lateral and vertical direction which results in X-ray response. The X-ray device a low detection limit and high SNR with good stability. However, there still have many questions about this MS which require major revision as below.

Response: Thanks very much for the reviewer's affirmation of our work. We have made revisions according to the great questions of the reviewer.

In Figure 2, the authors use DRS to characterize the bandgap for heterojunction Bi_xI_y materials which will benefit the e-h pair collection. However, why the heterojunction material have huge bandgap decrease com 1.67 eV to 0.7 eV is not clear discuss and understand in the main text.

Response: Good question, we made a mistake to take the bandgap of BiI (0.70 eV) as the bandgap of Bi_xI_y , which should be described as a dual bandgap. 1.67 eV is the bandgap of BiI_3 part and nearly not affected by the BiI (0.7 eV).

The corrected description in the revised manuscript is: "The reflectance of Bi_xI_y exhibits an indirect band nature with a sharp increase at 670-810 nm, assigned

to the thick BiI₃ layers, and a gentle rise after 810 nm, assigned to BiI layers. Namely, Bi_xI_y exhibits a dual bandgap. The dual bandgap of Bi_xI_y is also confirmed by the absorption spectrum (Fig. S8) of a typical grown Bi_xI_y. Based on the DRS result, the bandgap energy (E_g) of BiI layer was obtained by the Tauc method with 0.70 eV, dramatically lower than that of BiI₃ (1.67 eV) layer. The reduced bandgap of BiI promotes photo responses up to 1800 nm in the Bi_xI_y, as shown in Fig. S9.

Fig. 2b shows ultraviolet photoelectron spectroscopy (UPS) of the Bi_xI_y, which exhibits different cutoff energy (15.11 eV) and the energy (0.31 eV) extracted from linear extrapolation in the low-binding-energy region from which of BiI₃ (about 16.75 eV and 1.25 eV respectively, see ref. 32 and Fig. S4). As seen from Fig. 1f, the distance between BiI₃ and BiI layer (0.37 nm) is larger than which between BiI₃ layers (0.33 nm), resulting in rich of BiI in the stripped surface of Bi_xI_y. Therefore, the valence band energy (E_v) obtained by $E_v = 21.22 - 15.11 + 0.31 = 6.42$ eV is assigned to the BiI layer in the Bi_xI_y. The conduction band energy (E_c) of BiI layer is then calculated by $E_c = E_v + E_g = 5.72$ eV. The band diagram of Bi_xI_y heterostructure with a dual band gap can be plotted using the above data, as shown in Fig. 2c.”

From Figure 2c, the heterojunction band alignment for BiI₃ and Bi_xI_y indicates the CB different is much smaller than VB between Bi_xI_y and BiI₃ materials. However, when the e-h pairs generated by X-ray, the e-h pair separation flow in fig 1e is not match with Fig 2c and is confusing. Based on Band alignment in Fig 2c, how the e-h pairs generated and transport in the band alignment?

Response: Thanks for this great question. Yes, we made a mistake to describe the e-h pairs generation and transportation. Due to the band alignment in Fig 2c, the electron injection from BiI₃ layer to BiI layer would happen at the BiI₃-BiI interface, which generate a built-in electric field from BiI₃ to BiI. Therefore, the e-h pairs generated in the BiI₃-BiI interface under radiation would separate and flow in the BiI₃ (e) and BiI (h) layers respectively. The illumination of e-h pair separation has been corrected in Fig. 2e in the revised manuscript.

In Fig 2d, the authors used calculated $\mu\tau$ products to understand the charge transport between lateral and vertical. However, it might be due to the applied bias is not high enough for thick crystal (or vertical direction). I would suggest the author use SCLC measure with higher applied bias to further confirm the results.

Response: Thanks for this constructive suggestion. Yes, the saturation current is hardly achieved even the complete charge collection condition is met in a photoconductive detector with symmetry contacts. Therefore, the $\mu\tau$ products were recalculated by applying an asymmetry electrode configuration of Si/Bi_xI_y/Ag in the revised manuscript. To further understand the charge transport of Bi_xI_y heterostructure, we performed the SCLC measurements of Bi_xI_y with electrode configuration of Ag/Bi_xI_y/Ag. The results in Fig. S16 in the

revised manuscript exhibit a much larger mobility of $53 \text{ cm}^2 \text{ V}^{-1} \text{ s}^{-1}$ in the lateral direction than which in the vertical direction ($0.15 \text{ cm}^2 \text{ V}^{-1} \text{ s}^{-1}$), confirms the anisotropic charge transport in Bi_xI_y .

In Fig 3, the authors test the X-ray device performance at ambient condition. It's clear shows the difference of the X-ray response, SNR and detection limit for lateral and vertical devices. However, all of these device response time is in tens of seconds time regime. Why the device response time is so slow? The authors should have a detail discuss and comparison with reported other X-ray material candidates.

Response: Thanks for this constructive suggestion. We gave detailed on/off responses of Bi_xI_y in a millisecond time regime, as shown in Fig. S13 in the revised manuscript. We performed the comparison of Bi_xI_y with other x-ray material in the revised manuscript as: A detailed on/off response of Bi_xI_y (Fig. S13) shows rise/fall times of 49/71 ms for lateral device and 74/98 ms for vertical device under 1 V mm^{-1} bias, which is comparable to the single crystal BiI_3 (110/120 ms) grown by physical vapor transport (PVT)¹⁴ and the printable MAPbI_3 device (50 ms) for imaging³⁴. From the comparison results we can see that the response time of Bi_xI_y is not very slow.

From fig 4f, the device response before and after water immersion is very encouraging, however, the crystal color changed after water immersion (or hydrolysis) and not quite the same in fig 1b. The authors should carefully characterized the material before and after water immersion in order to claim the material is not effected by water.

Response: Thanks for this constructive suggestion. Yes, Bi_xI_y is very stable for moisture. However, we cannot claim it is not affected by water. We added water immersion results (Fig. 4f and Fig.S18 in the revised manuscript) up to 8 hours. It can be seen from Fig. S18 that the color of Bi_xI_y was changed after 8 hours immersion. However, after peeled off the corrupted surface, we can see that the interior of Bi_xI_y remained unchanged. In a word, water only affected the surface of Bi_xI_y . A nearly unchanged device response after 8 hours immersion is observed, as shown in Fig. 4f.

Also, from many device response, why some of the devices have baseline drifting issue and some not, why does these happen to the same material and devices?

Response: Thanks for the question. Yes, devices with coplanar electrode configuration (Fig. S9 in the revised manuscript) used for infrared or visible light detection would exhibit baseline drifting, which may due to the surface defects such as layer distortion, chipping and crack caused by stripping, surface oxidation, moisture and so on. However, the baseline is stable in the same material and devices for x-ray detection with vertical or lateral electrode configurations (Fig. 3a).

Some minor correction:

(1) Some fig labeling and ordering in the main text is wrong, such as page 9 line 188. The fig 2e should be fig 2c. Fig2d is after fig 2e. Same for Fig 3. Please correct these issue through the whole MS.

Response: Yes, it has been corrected in the **revised manuscript**.

(2) In Fig 2d, what does the parallel and vertical E means? It's confusing and not clear to reader. I would suggest the authors add a cartoon in the Fig 2d and clearly describe in the MS.

Response: Yes, we replaced the $E//c$ as vertical and $E\perp c$ as lateral in all the Figures in the **revised manuscript** to avoid the confusion. Vertical and lateral configuration are shown in Fig. 3a.

(3) Fig 4c and 4e are missing the right axis label

Response: Yes, the right axis label is added **in Fig. 4c in the revised manuscript**. Only the upper of Fig. 4e need a right axis label of '**SNR**', the lower of which need not.

Reviewer #3:

As the first comment, I have to acknowledge the idea of BiI/BiI₃/BiI layered sample for use as an X-ray detector.

Besides this interesting idea, the characterization is lacking seriously.

Besides concerns on the electrical characterization, I encourage the authors to use neutral scientific language.

I.e champion material, ultra-high....are used everywhere. I know, these days, it is common to use superlatives in a scientific text, but this is a really bad practice. We have to show good scientific results - we are not selling on a marketplace.....

For the electrical characterization, I see several contradictions:

Response: Thanks very much for the reviewer's affirmation of our work. Yes, we used neutral scientific language in the **revised manuscript**. Also, we have made corrected characterizations according to the great comments of the reviewer.

*) the authors report high resistivities for 0.7eV Bandgap material?! These high values are unphysical and are most likely caused by bad electrical contacts - not surprising for a Cu tape as a contact.

Response: Thanks very much for the great question. We made a mistake to

take the bandgap of BiI (0.70 eV) as the bandgap of Bi_xI_y, which should be described as a dual bandgap. The resistivity of Bi_xI_y is determined by the main BiI₃ part with a medium bandgap (1.67eV). Yes, Cu tape has a great impact on resistivity. Therefore, we used a Ag/Bi_xI_y/Ag configuration with a probe to retest the resistivities of Bi_xI_y and achieved values of $4.1 \times 10^9 \Omega \text{ cm}$ for lateral device and $2.2 \times 10^{10} \Omega \text{ cm}$ for vertical device respectively, as shown in **Fig. 2d in the revised manuscript**. The resistivities of Bi_xI_y are similar to which of BiI₃ (10^8 - $10^{13} \Omega \text{ cm}$).

*) the authors report on extremely high sensitivities. What you see here is photoconductive gain. Very typical for a symmetric device. The IV of such a device has to be linear (not seen in the semi-log plot in Fig 2d). For such devices, you can get any sensitivity by applying a higher bias.

Response: Good question, according to Hecht theory, the photocurrent would increase nonlinear as the bias increase and reach a saturated value at a sufficient high bias under radiation due to the complete collection of the generated charge. However, it is hardly to saturate in a photoconductive detector with a symmetric electrode configuration because the photocurrent would still increase as the bias exceed the complete charge collection value in this type of detector. Therefore, we see the nonlinear increase but unsaturated photocurrent in the Fig. 2d. The nonlinear increase but unsaturated photocurrents are also observed in many works using a photoconductive detector with a symmetric electrode (commonly Au/single crystal/Au). The high sensitivities of our photoconductive detector under high bias (up to 200 V) indicate the material could apply a relatively high working bias. We know it is not seriously to use the highest value as the sensitivity of photoconductive detector. Therefore, by considering the largest SNR of our detectors at approx. 50 V bias, we used the sensitivity under 50 V to compare with other detectors in the **revised manuscript**.

*) $\mu\tau$ product cannot be obtained from a symmetric device. I saw this approach in many papers but is an incorrect interpretation. The $\mu\tau E$ product defines the Schubweg of the charge carriers. At a certain bias, the Schubweg equals the sample thickness. For such an experiment you need ASYMMETRIC contacts (or at least one blocking). For examples. Take as reference a commercial CdTe photodiode. There you'll see sensitivity in the range of 3000-400 $\mu\text{C}/\text{Gscm}^2$ clearly saturating and exactly fitting the theoretical limit given by the Klein rule! Every value higher than this points to a leaky diode or photoconductive gain. Sorry to say, your treatment is not correct. I'm picky here, as during my time in the industry I had to do such tests regularly.

Response: Thanks for the constructive suggestion. Yes, it is not seriously to calculate $\mu \tau$ product by using a photoconductive detector with symmetric electrode. Therefore, we recalculated the $\mu \tau$ products with an asymmetric

device configuration of Si/Bi_xI_y/Ag, in which saturated photocurrents were achieved. The new results were presented in Fig. S15 in the revised manuscript.

*) Detection limit. Has to be measured at the same bias as the photocurrent. You cannot compare a 0V with the PC under bias. The correct treatment is to use a spectrum analyzer and determine the spectral noise density and from that, you can calculate the SNR.

Response: Yes, a single heterojunction of BiI/BiI₃ may be attractive for self-power detection, however which cannot be afforded by us at present. It really doesn't make sense to use a bulk Bi_xI_y for self-power detection. Therefore, we deleted relevant descriptions in the revised manuscript.

In summary. You are showing a highly interesting material system, but the characterization as an X-ray detector is lacking seriously.

Response: Thanks very much for the comment, the characterization has been corrected in the revised manuscript.

Reviewer #4:

In the current work, Zhuang R. et. al demonstrate the synthesis of novel heterostructure Bi_xI_y, which consists of sequential layers of BiI₃ and BiI, and based on that heterostructure the X-ray detector prototype. The authors did impressive work on the synthesis and the structural characterization of the Bi_xI_y material, clearly demonstrating its heterostructure form and well describing the novel solution-grown process. Optical and X-ray detection characterization was done partially correctly, proving high sensitivity to X-ray, low lowest detection limit, and reasonable stability. Overall the proposed Bi_xI_y heterostructure seems to be a promising material for X-ray detection. However, there are several major concerns, if unresolved, questioning the value of this work. The first concern is whether Bi_xI_y heterostructure is much better in terms of X-ray detection properties than a simple BiI₃ single crystal, which is pretty known in the literature as a good gamma and an X-ray detector, as the authors cite in ref 22-36. The second problem is the correctness of the Bi_xI_y heterostructure bandgap determination. Thus, I would recommend this manuscript for publication in Nature Communication only when all major and minor concerns are addressed, which are listed in detail below.

Response: Thanks very much for the reviewer's great comments, which point out two major deficiencies in our work. The Bi_xI_y heterostructure we reported here is not a defect of BiI₃ single crystal or a simply mix of BiI₃ and BiI. Our work proposed a new type of bulk van der Waals heterostructure prepared by cheap and environment-friendly solution method for x-ray detection. The method could be extended to prepare other bulk van der Waals heterostructure such as 2D perovskite-based materials, as shown below. In a word, the main

purpose of this work is not to perform a comprehensive comparison between Bi_xI_y and BiI_3 single crystal. However, we agree that a direct comparison of X-ray detection performance between Bi_xI_y and BiI_3 single crystal is still necessary to clarify whether heterostructure has advantages (and why) or not.

We have made corrections in the **revised manuscript** according to the concerns of the reviewer.

Bismuth based 2D perovskite heterostructure and its diffuse reflectance spectrum exhibits a dual bandgap.

Major remarks

1. For a clear demonstration of this work advance, absolutely necessary is the direct comparison of X-ray detection performance between the fabricated Bi_xI_y heterostructure and similar size BiI_3 single crystal. While the authors provide in Supplementary Information a comparison of X-ray sensitivities (Fig. 15S), noting the sensitivity of BiI_3 , it is hard to compare this value with the current work, since for BiI_3 reference isn't even stated. For a fair comparison, I suggest to the authors synthesize their own BiI_3 SC or utilize commercial ones, which the authors already used in work, fabricate the detector devices, and perform similar X-ray detection characterization. It is important to compare resistivity, mobility-lifetime, X-ray sensitivity, and lowest detection limit, to prove (or disprove) the advances achieved by the Bi_xI_y heterostructure. In the current state of the manuscript, only resistivity is compared properly (it is similar to the fabricated Bi_xI_y heterostructure and previously reported BiI_3), which definitely isn't a comprehensive comparison.

Response: Thanks for the constructive suggestion. We tried to grow BiI_3 single crystal from solution, however only powder precipitants were obtained. The method of growing BiI_3 single crystal from solution is also not reported until now. Our commercial BiI_3 is powders and can not be used to fabricate single

crystal device. Therefore, we compared the Bi_xI_y heterostructure with the recent reported BiI_3 single crystal grown by PVT (ref. 19 in the revised manuscript), which has similar device configurations and size (several millimeters) in the revised manuscript as: "The sensitivities of Bi_xI_y detectors are shown in Fig. 4d. The lateral device had a much larger photocurrent density than the vertical device and achieved a high sensitivity of $4.3 \times 10^4 \mu\text{C Gy}^{-1} \text{cm}^{-2}$ at 24 V mm^{-1} (50 V bias), which is comparable to the newly reported high-sensitive detectors shown in Fig. S17. Moreover, the sensitivity of lateral device achieves nearly one order of magnitude higher than that of the lateral BiI_3 detector ($0.5 \times 10^4 \mu\text{C Gy}^{-1} \text{cm}^{-2}$ at 20 V mm^{-1})¹⁹, confirmed the advantage of heterostructure of Bi_xI_y for x-ray detection. However, the sensitivity of vertical device ($253 \mu\text{C Gy}^{-1} \text{cm}^{-2}$ at 19.5 V mm^{-1}) is smaller than which of the vertical BiI_3 single crystal detector ($660 \mu\text{C Gy}^{-1} \text{cm}^{-2}$ at 20 V mm^{-1})¹⁹. A larger distance between BiI_3 and BiI layers than which between BiI_3 layers leads to easier mechanical exfoliation of Bi_xI_y , however harms the charge transport and then reduce the sensitivity of the vertical detector."

2. In the manuscript the only major physical properties, which is shown to be sufficiently different from BiI_3 , is the bandgap. While the authors stated correctly, that a lower bandgap is beneficial for X-ray detection in terms of lower e-h creation energy, the bandgap values seem to be only estimated for BiI part, but not the whole heterostructure. For correct bandgap estimation of mixture compound, please look (DOI: 10.1021/acs.jpcclett.8b02892). It looks like there are separate bandgap values for different parts of the heterostructure for BiI and BiI_3 parts, and the last isn't changed compared to commercial BiI_3 . As the authors stated, BiI contribution to chemical composition is tiny compared to BiI_3 . Thus, X-ray are mostly attenuation in BiI_3 layers, and the effect of the low BiI bandgap on the e-h pairs creation energy is doubtful. This statement should be either clarified or removed from the manuscript. Despite that, I still think that heterostructure could have beneficial for charge transport properties compared to BiI_3 , which is still to be proved, as it is stated in 1st comment.

Response: Thanks for the great comment. We made a mistake to take the bandgap of BiI (0.70 eV) as the bandgap of Bi_xI_y . However, unlike $\text{TiO}_2 + \text{MO}$ reported in DOI: 10.1021/acs.jpcclett.8b02892, Bi_xI_y heterostructure is not a simply mix of BiI_3 and BiI , which has only one melting point slightly different from BiI_3 , as seen from the DSC measurement (Fig. S3 in the revised manuscript). Therefore, taking the bandgap of BiI_3 (1.67 eV) as the bandgap of Bi_xI_y is also incorrect. Bi_xI_y should be described as a dual bandgap. We agree that the minor part BiI contribute negligible for the e-h pairs creation although its bandgap is very small and removed the related descriptions in the revised manuscript. TA measurement shows a reduced lifetime of the excited electron in the BiI_3 layers, which indicates the charge separation and transportation in the BiI_3 - BiI interface should plays an important role in the high x-ray detection performance although x-ray is mostly attenuation in BiI_3 layers. The lateral μ

τ product of Bi_xI_y exhibits a much larger value ($3.0 \times 10^{-3} \text{ cm}^2 \text{ V}^{-1}$, lateral) than the reported value ($3.4 \sim 9.5 \times 10^{-6} \text{ cm}^2 \text{ V}^{-1}$) of BiI_3 single crystal, which proves the heterostructure of Bi_xI_y is beneficial for the charge collection.

Minor remarks

1. As being mentioned above, BiI_3 material is well-known for radiation detection. While the authors mentioned a lot of perovskite references in the introduction, BiI_3 research works are mentioned only in the context of resistivity. I suggest adding references on BiI_3 detectors in the introduction, stating that this work is advanced compared to other BiI_3 reports.

Response: Thanks for the constructive suggestion. The references on BiI_3 detectors have been added in the **revised manuscript**.

2. Please add the caption for Figure 1f.

Response: Yes, it has been corrected in the **revised manuscript**.

3. I wouldn't use a double axis for Figure 4c, since it makes it difficult to compare two different directions. Instead, I would suggest making one axis with a log scale.

Response: Yes, it has been corrected in the **revised manuscript**.

4. In the inset of Figure 2a, where the authors report the bandgap for Bi_xI_y (probably to BiI , as mentioned above), the authors may consider changing the Y-axis power from $1/2$ to 2, since the first is used for direct semiconductors, while the last is for indirect one. From reflectance dependence on main Figure 2a, it seems that BiI part has indirect semiconductor behavior.

Response: For Tauc plot, The relation between the absorption coefficient α and incident photon energy $h\nu$ near the fundamental absorption edge is:

$$(\alpha h\nu)^{1/n} = A(h\nu - E_g).$$

The exponent n depends on the type of optical transition. For direct transition, $n = 1/2$; for **indirect transition, $n = 2$** (ref. 12 in the revised manuscript).

For the reflectance, there is Kubelka-Munk equation:

$$F(R) = \alpha/S = (1-R)^2/2R.$$

where α and S are the absorption and scattering coefficients, respectively, and R is the reflectance. S is assumed to be constant with wavelength for the same sample condition.

Therefore, we could obtain: $(F(R)h\nu)^{1/2} = A(h\nu - E_g)$ for **indirect transition**.

All changes are marked by the red color.

REVIEWER COMMENTS

Reviewer #2 (Remarks to the Author):

The authors have answered most of my questions and the revised MS have improved.

However, I still have some minor comments on the response.

(1) For the device response time. it seems like all the single crystal response time either in BiI₃, perovskite and material from this MS are all in MS regime, and some of the reported perovskite thin film devices have device response in μs . This suggest that it might due to the applied field is not strong enough that results in slow response.

(2) For Fig 4b, it's better to show the complete photon energy range up to 1000 keV for reader's convenience.

(3) In Fig 4F, the author claim it's unchanged after 8 hours of water immersions for the X-ray device response, however, the X-ray response peak shape changed for the immersion device which it normally can be observed in degraded or partially damaged device. This is contradict to what they have concluded in the MS. Therefore, I would recommend the authors to revise their discussion in the MS.

Reviewer #4 (Remarks to the Author):

In the revised manuscript, Zhuang R. et. al significantly improved the quality of the manuscript, addressing most of my comments. Now the authors have clarified the confusion with the bandgap estimation and added the comprehensive comparison of the BixIy heterostructure with previous BiI₃ reports, including the sensitivity and the charge transport properties. I find it especially prominent that the authors demonstrate the enhanced mobility-lifetime product of a later asymmetrically contacted BixIy device. It is higher on several orders of magnitude than the one obtained with previously reported BiI₃ single crystals. I believe that this might be the main advance of the current work, which demonstrates clearly why the BixIy heterostructure enables prominent X-ray detection performance. However, to have solid evidence of such significant enhancement, I suggest for the authors perform a few additional comprehensive characterization experiments, in detail described below. Once the authors address my major comment on this matter and one other minor comment, I will feel confident to recommend this manuscript for publication in Nature Communication.

Major comment

The Hecht equation fit of the photocurrent vs bias voltage and SCLC aren't sufficient enough to confidently verify the high mobility-lifetime product, since those methods for some materials suffer from voltage depended on artifacts, leading to overestimation of mobility-lifetime product. Thus, for comprehensive characterization of the BixIy heterostructure charge transport, I found it necessary, to perform independent measurements of mobility and lifetime. For the mobility estimation, I suggest utilizing either the Time-of-Flight technique or the Hall effect, whatever the authors find more feasible for BixIy heterostructure. For the lifetime measurements, I would invite the authors to utilize either the microwave photoconductivity decay method or the transient photocurrent technique. It would be interesting to compare the value of the mobility-lifetime product obtained by additional independent measurements with the value, already demonstrated in the manuscript.

Minor comment

Line 242-243 in the manuscript - "... the printable MAPbI₃ device (50 ms) for imaging³⁴".

I don't agree that the transient response in Ref³⁴ is equal to 50 ms. Looking carefully at Fig.4d from Ref³⁴, the transient response is faster than 50 ms, since the reported detector is able to

distinguish clearly the pulses with a width of 50 ms. The actual rise time is much faster (although fall time indeed is comparable with 50 ms). To be more accurate on this matter, I would suggest the authors change their statement to "... the printable MAPbI₃ device (less than 50 ms) for imaging³⁴".

Replies to the comments of reviewer #2:

The authors have answered most of my questions and the revised MS have improved.

However, I still have some minor comments on the response.

Response: Thanks very much for the reviewer's comments. We have made further revisions according to the additional comments of the reviewer.

(1) For the device response time. it seems like all the single crystal response time either in BiI₃, perovskite and material from this MS are all in MS regime, and some of the reported perovskite thin film devices have device response in μ s. This suggest that it might due to the applied field is not strong enough that results in slow response.

Response: Yes, a higher applied field would result in a faster response. Due to its high SNR, Bi_xI_y could employ a relatively high bias (> 50 V) to accelerate its response. However, our devices response time derived from the on/off measurements are governed by the time for x-ray tube current changing from zero to maximum or vice versa and the minimum time interval between two adjacent counting points of source meter, all in MS regime. Therefore, the response time values of our device are conservative. The response of Bi_xI_y devices are still faster than which of BiI₃ single crystal (also tested by on/off measurement, ref. 14) even under a much lower bias, indicating there don't exit large number of trap states in the heterostructure which may slow down the response. We know that some of the reported perovskite thin film devices have device response in μ s. The perovskite device we selected to compare with Bi_xI_y has a similar thickness of 0.83 mm.

(2) For Fig 4b, it's better to show the complete photon energy range up to 1000 keV for reader's convenience.

Response: The photon energy range in Fig 4b has been extended to 1000 KeV in **the revised manuscript**.

(3) In Fig 4F, the author claim it's unchanged after 8 hours of water immersions for the X-ray device response, however, the X-ray response peak shape changed for the immersion device which it normally can be observed in degraded or partially damaged device. This is contradict to what they have concluded in the MS. Therefore, I would recommend the authors to revise their discussion in the MS.

Response: Thanks for the question. The Bi_xI_y specimens after water immersion were dried by handkerchief tissues first and then repasted the Cu tapes for the following I-V tests. The discrepancy of the contact between Cu tape and Bi_xI_y before and after water immersion would lead to the slight changing of X-ray response peak shape, but hardly affect the response photocurrent intensity. For a more accurate description, We have revised the discussion in **the revised**

manuscript as: “Moreover, nearly unchanged photocurrent intensity (Fig 4f) of the Bi_xI_y detectors could be observed even after 8 h water (20°C) immersion (Fig. S18), confirms its superior environmental stability.”

Replies to the comments of reviewer #4:

In the revised manuscript, Zhuang R. et. al significantly improved the quality of the manuscript, addressing most of my comments. Now the authors have clarified the confusion with the bandgap estimation and added the comprehensive comparison of the Bi_xI_y heterostructure with previous BiI₃ reports, including the sensitivity and the charge transport properties. I find it especially prominent that the authors demonstrate the enhanced mobility-lifetime product of a later asymmetrically contacted Bi_xI_y device. It is higher on several orders of magnitude than the one obtained with previously reported BiI₃ single crystals. I believe that this might be the main advance of the current work, which demonstrates clearly why the Bi_xI_y heterostructure enables prominent X-ray detection performance. However, to have solid evidence of such significant enhancement, I suggest for the authors perform a few additional comprehensive characterization experiments, in detail described below. Once the authors address my major comment on this matter and one other minor comment, I will feel confident to recommend this manuscript for publication in Nature Communication.

Response: Thanks very much for the reviewer’s great comments. We found from the reviewer’s comment that we made a wrong comparison between the lateral Bi_xI_y device and the previous reported BiI₃ single crystal devices, which used a vertical contact configuration, as shown below. This comparison is meaningless and leads to a wrong conclusion that the mobility-lifetime product of Bi_xI_y is higher on several orders of magnitude than which of previously reported BiI₃ single crystal. Considering the deviation caused by different test methods and sample size etc., the mobility-lifetime product of the vertical Bi_xI_y device is comparable to which of the vertical BiI₃ device. However, we feel sorry that we didn’t find the mobility-lifetime product of the lateral BiI₃ single device from the previous reported works. Therefore, we corrected the relevant description in the revised manuscript.

BiI₃ single crystal device with vertical Au contacts from ref 13. BiI₃ grown by PVT is usually very thin and hardly to construct a lateral device.

Major comment

The Hecht equation fit of the photocurrent vs bias voltage and SCLC aren't sufficient enough to confidently verify the high mobility-lifetime product, since those methods for some materials suffer from voltage depended on artifacts, leading to overestimation of mobility-lifetime product. Thus, for comprehensive characterization of the Bi_xI_y heterostructure charge transport, I found it necessary, to perform independent measurements of mobility and lifetime. For the mobility estimation, I suggest utilizing either the Time-of-Flight technique or the Hall effect, whatever the authors find more feasible for Bi_xI_y heterostructure. For the lifetime measurements, I would invite the authors to utilize either the microwave photoconductivity decay method or the transient photocurrent technique. It would be interesting to compare the value of the mobility-lifetime product obtained by additional independent measurements with the value, already demonstrated in the manuscript.

Response: Thanks for the comment. A reliable lifetime estimation of Bi_xI_y heterostructure should consider the lifetime of transition in BiI_3 layer, BiI layer and between BiI_3 - BiI interface. TA measurement shows a reduced lifetime of the excited electron in the BiI_3 layers. However, we failed to obtain fluorescence emission of Bi_xI_y at 1000 ~ 2000 nm which corresponding to the BiI or BiI_3 - BiI interface emission and therefore could not afford the lifetime because the quantity of BiI in Bi_xI_y is too small. Therefore, we used the Hecht equation to estimate the mobility-lifetime product of Bi_xI_y , which is a general method at present. We made a wrong comparison between the lateral Bi_xI_y device and the previous reported vertical BiI_3 single crystal devices and drew a wrong conclusion that the mobility-lifetime product of Bi_xI_y is higher on several orders of magnitude than BiI_3 . However, by roughly comparing the mobility-lifetime products tested by the same method and sensitivities of Bi_xI_y ($3.0 \times 10^{-3} \text{ cm}^2 \text{ V}^{-1}$, $4.3 \times 10^4 \text{ uC Gy}^{-1} \text{ cm}^{-2}$ at 24 V mm^{-1} , lateral) and $\text{Cs}_2\text{AgBiBr}_6$ ($6.3 \times 10^{-3} \text{ cm}^2 \text{ V}^{-1}$, $105 \text{ uC Gy}^{-1} \text{ cm}^{-2}$ at 25 V mm^{-1} , ref. 26) and $\text{Cs}_3\text{Bi}_2\text{I}_9$ ($7.97 \times 10^{-4} \text{ cm}^2 \text{ V}^{-1}$, $1652.3 \text{ uC Gy}^{-1} \text{ cm}^{-2}$ at 50 V mm^{-1} , ref. 42), we think the mobility-lifetime product of Bi_xI_y at lateral direction is not obvious overestimation. The barriers derived from contacts have a trivial impact in mobility-lifetime product estimation.

Minor remarks

Line 242-243 in the manuscript -"... the printable MAPbI_3 device (50 ms) for imaging³⁴".

I don't agree that the transient response in Ref34 is equal to 50 ms. Looking carefully at Fig.4d from Ref34, the transient response is faster than 50 ms, since the reported detector is able to distinguish clearly the pulses with a width of 50 ms. The actual rise time is much faster (although fall time indeed is comparable with 50 ms). To be more accurate on this matter, I would suggest the authors change their statement to "... the printable MAPbI3 device (less than 50 ms) for imaging34".

Response: Yes, it has been corrected in **the revised manuscript**.

All changes are marked by the **red color**.

REVIEWERS' COMMENTS

Reviewer #2 (Remarks to the Author):

The authors have answered all my comments and concerns. I would suggest publishing this MS for Nature communications.

Reviewer #4 (Remarks to the Author):

In the revised manuscript Zhuang R. et al. in principle answered all my comments. Now the authors highlight properly in the main text achievements and provide a comprehensive material characterization description. The only minor comment – I suggest to the authors add more of their main advances in the abstract: demonstration of solution-grown crystal BiI₃ with good charge transport and thickness, suitable for X-ray absorption. In previous reports, large crystals of similar BiI₃ were only achieved with melt techniques. I believe the major advance of this manuscript in the X-ray detectors materials field is the good X-ray detection properties (including high X-ray sensitivity, mobility-lifetime product, size, and resistivity) obtained with solution-grown crystals from the lead-free and not-perovskite material. Thus, all of this should be highlighted in the abstract. Except of that, I believe that the manuscript is well-written, the data supports clearly the main claims and the scientific advance in the X-ray detection field is significant. Therefore, I would recommend this manuscript for publication in Nature Communication, after addressing my only minor comment mentioned above.

Replies to the comments of reviewer #4:

In the revised manuscript Zhuang R. et al. in principle answered all my comments. Now the authors highlight properly in the main text achievements and provide a comprehensive material characterization description. The only minor comment – I suggest to the authors add more of their main advances in the abstract: demonstration of solution-grown crystal Bi_xI_y with good charge transport and thickness, suitable for X-ray absorption. In previous reports, large crystals of similar BiI_3 were only achieved with melt techniques. I believe the major advance of this manuscript in the X-ray detectors materials field is the good X-ray detection properties (including high X-ray sensitivity, mobility-lifetime product, size, and resistivity) obtained with solution-grown crystals from the lead-free and not-perovskite material. Thus, all of this should be highlighted in the abstract. Except of that, I believe that the manuscript is well-written, the data supports clearly the main claims and the scientific advance in the X-ray detection field is significant. Therefore, I would recommend this manuscript for publication in Nature Communication, after addressing my only minor comment mentioned above.

Response: Thanks very much for the reviewer's recognition of our work. According to the reviewer's great comment, the abstract has been revised **in the revision manuscript as:** " X-ray detectors must be operated at minimal doses to reduce radiation health risks during X-ray security examination or medical inspection, therefore requiring high sensitivity and low detection limits. Although organolead trihalide perovskites have rapidly emerged as promising candidates for X-ray detection due to their low cost and remarkable performance, these materials threaten the safety of the human body and environment due to the presence of lead. Here we demonstrate the realization of highly sensitive X-ray detectors based on an environmentally friendly solution-grown thick $\text{BiI}/\text{BiI}_3/\text{BiI}$ (Bi_xI_y) van der Waals heterostructure. The devices exhibit anisotropic X-ray detection response with a sensitivity up to $4.3 \times 10^4 \mu\text{C Gy}^{-1} \text{cm}^{-2}$ and a detection limit as low as 34 nGy s^{-1} . Meanwhile, the obtained Bi_xI_y detectors also exhibit high environmental and hard radiation stabilities. Our work motivates the search for new van der Waals heterostructure classes to realize high-performance X-ray detectors and other optoelectronic devices without employing toxic elements."

All changes are marked by the red color.